# Normal tissue architecture determines the evolutionary course of cancer

Jeffrey West [1✉], Ryan O. Schenck [1,2], Chandler Gatenbee [1], Mark Robertson-Tessi [1] & Alexander R. A. Anderson [1✉]

Cancer growth can be described as a caricature of the renewal process of the tissue of origin, where the tissue architecture has a strong influence on the evolutionary dynamics within the tumor. Using a classic, well-studied model of tumor evolution (a passenger-driver mutation model) we systematically alter spatial constraints and cell mixing rates to show how tissue structure influences functional (driver) mutations and genetic heterogeneity over time. This approach explores a key mechanism behind both inter-patient and intratumoral tumor heterogeneity: competition for space. Time-varying competition leads to an emergent transition from Darwinian premalignant growth to subsequent invasive neutral tumor growth. Initial spatial constraints determine the emergent mode of evolution (Darwinian to neutral) without a change in cell-specific mutation rate or fitness effects. Driver acquisition during the Darwinian precancerous stage may be modulated en route to neutral evolution by the combination of two factors: spatial constraints and limited cellular mixing. These two factors occur naturally in ductal carcinomas, where the branching topology of the ductal network dictates spatial constraints and mixing rates.

[1] Integrated Mathematical Oncology Department, H. Lee Moffitt Cancer Center & Research Institute, Tampa, FL, USA. [2] Wellcome Centre for Human Genetics, University of Oxford, Oxford, UK. ✉email: jeffrey.west@moffitt.org; Alexander.Anderson@moffitt.org

Cancer has been hypothesized to be a caricature of the renewal process of the tissue of origin: arising from (and maintained by) small subpopulations capable of continuous growth[1]. The strong influence of the tissue structure has been convincingly demonstrated in intestinal cancers where adenomas grow by the fission of stem-cell-maintained glands influenced by early expression of abnormal cell mobility in cancer progenitors[2,3]. So-called "born to be bad" tumors arise from progenitors which may already possess the necessary driver mutations for malignancy[4,5] and metastasis[6]. These tumors subsequently evolve neutrally, thereby maximizing intratumoral heterogeneity and increasing the probability of therapeutic resistance.

The heterogeneous collection of tens of thousands of somatic alterations may be classified into drivers (conferring advantageous, cancerous phenotypes to neoplastic cells) and passengers (neutral, nearly-neutral, or slightly deleterious mutations). Highly deleterious mutations are subject to negative selection and are removed from the population, while moderately deleterious mutations can evade purifying selection to remain present in an evolving tumor under selection pressure, a process known as "hitchhiking" with the sweeping driver clone[7,8].

Patterns of intratumoral genetic heterogeneity (ITH) and subclonal architecture are the direct consequence of the evolutionary dynamics of tumor growth. Multiregion sequencing has produced evidence that Darwinian evolution shapes at least part of ITH[9,10]. Substantial increases in subclone fitness have been observed in some cases: 21% of colon cancers, 29% of gastric cancers, and 53% of metastases[11].

Several evolutionary models have been proposed to describe the transition from premalignant to invasive tumor growth[12]. For example, several studies indicated that cell lineages for ductal carcinoma in situ (DCIS) tumors were distinct from invasive ductal carcinoma (IDC) tumors[13,14]. This gives rise to an 'independent lineage model' which holds that there are different initiating cells for in situ and invasive populations, respectively[15]. In contrast, others have proposed an 'evolutionary bottleneck' model whereby multiple clones evolve within the ducts, overcoming obstacles such as spatial constraints, nutritional limitation, and immune attack[16]. The parental clone that can conquer these obstacles emerges from the subclones generated by early Darwinian evolution. This evolutionary bottleneck establishes key driver mutations in the invasive parental clone, which are the ubiquitous mutations measured during the neutral phase of tumor growth[17,18]. Casasent et al. challenged both evolutionary paradigms using single-cell DNA sequencing to measure copy number profiles of single tumor cells while preserving spatial information in synchronous DCIS-IDC patients[19,20]. These data show that multiple subclones which evolved in ducts during in situ stages subsequently co-migrated to surrounding tissues, indicating a high degree of overlap in heterogeneity within and outside the ducts.

How does tissue structure influence somatic evolution? Some have speculated that modes of evolution (Darwinian to neutral) may be the outcome of cellular architecture of the tumor (e.g., the glandular structure of colorectal cancers or the ductal structure of ductal carcinomas can limit the effects of selection), or of the malignancy's anatomical location, governing access to resources, or strong spatial constraints for growth[5,11,21]. Some structures are "amplifiers" of natural selection, improving the odds of advantageous mutants[22,23] (e.g., pancreatic ducts that control migration rates between patches[24] and spatially segregated colon glands with a centralized stem cell pool[3]). The well-defined tissue structure begins to break down as the cancer transitions to increasing invasiveness, often with sudden, "punctuated" accrual of copy number alterations needed to facilitate invasion into the stroma[25,26]. It is our point of view that this switch from Darwinian to neutral evolution is highly influenced by the tissue structure where the founding clone arises, and that transition to neutrality may occur early in tumor progression (before invasion).

The effect of spatial structure, dispersal, migration, and tissue turnover is well-studied topics in population ecology and population genetics. Spatial structures allow for the emergence of coexistence between populations with differential fitness[27] even when birth and death are spatially decoupled[28]. Hierarchically organized structures in self-renewing tissues limit the accumulation of somatic mutations[29]. Cellular differentiation "washes" out harmful mutations, while hierarchical architecture limits overall cellular turnover required to maintain the tissue, thus limiting the chance for new mutations[30]. The probability of fixation of an allele is strongly position-dependent: alleles near centrally-located, high turnover regions are orders of magnitude more likely to fix[31]. Barton et al. showed that the slow-down of clonal sweeping caused by large domains tends also to reduce the size of the genomic region over which diversity is depressed by a sweep[32]. The role of dispersal between spatially segregated habitats or "patches", is also well-studied, often utilizing multi-path metapopulation models under weak-selection[33]. Migrative potential is shown to amplify the invasion probability of a heterogeneous population[34], an effect that increases when the dispersal network is more structured[23].

While many mathematical passenger–driver tumor evolution models use branching processes[35] or stochastic, well-mixed (non-spatial), agent-based models[21,36–39], some have investigated the role of spatial competition on local heterogeneity and circulating tumor cells[40], resistance to therapy[41], metastasis[6,27], and trade-offs between migration and proliferation[42]. The model introduced here is a spatially-explicit extension of a previously published non-spatial model of passengers and drivers in tumor evolution[7,36,37].

In this work, we extend these findings on the importance of structure, dispersal, migration, and turnover to a more biologically realistic setting: the three-dimensional branching topology of a breast ductal network spatial structure, recapitulating the intratumoral heterogeneity in precancerous lesions of ductal carcinoma in situ (DCIS). In this setting, two otherwise identical tumors may realize dramatic differences in the fitness depending on constraints imposed by tissue architecture. This leads us to the following insight: the surrounding spatial context modulates the "realized tumor fitness", defined as the rate of change of the ratio of driver mutations to mutation burden.

## Results

Tumor evolution is played out on a two- or three-dimensional grid where each grid point can contain at most one cell. Each cell carries heritable genetic changes classified into driver mutations (e.g., an activating mutation in KRAS) or passenger mutations. Cells begin each simulation with a single driver mutation, where each subsequent driver increases the birth rate (i.e., multiplicative epistasis) by a factor of a fitness advantage parameter, $s_d$. Similarly, passenger mutations decrease the birth rate by a factor of fitness penalty, $s_p$. In Fig. 1, we quantify heterogeneity of both drivers and passengers. Results for neutral and deleterious passengers across several diversity metrics are shown in the supplementary information (see Supplementary Figs. 1–4). $T_{p/d}$ is the mutation target sizes for drivers/passengers[36], such that the effective mutation rate is given by $\mu_{p/d} = \mu T_{p/d}$. We begin with a systematic and generalized understanding of the underlying mathematical model (with varied spatial domain sizes and mixing rates) in Figs. 1 and 2.

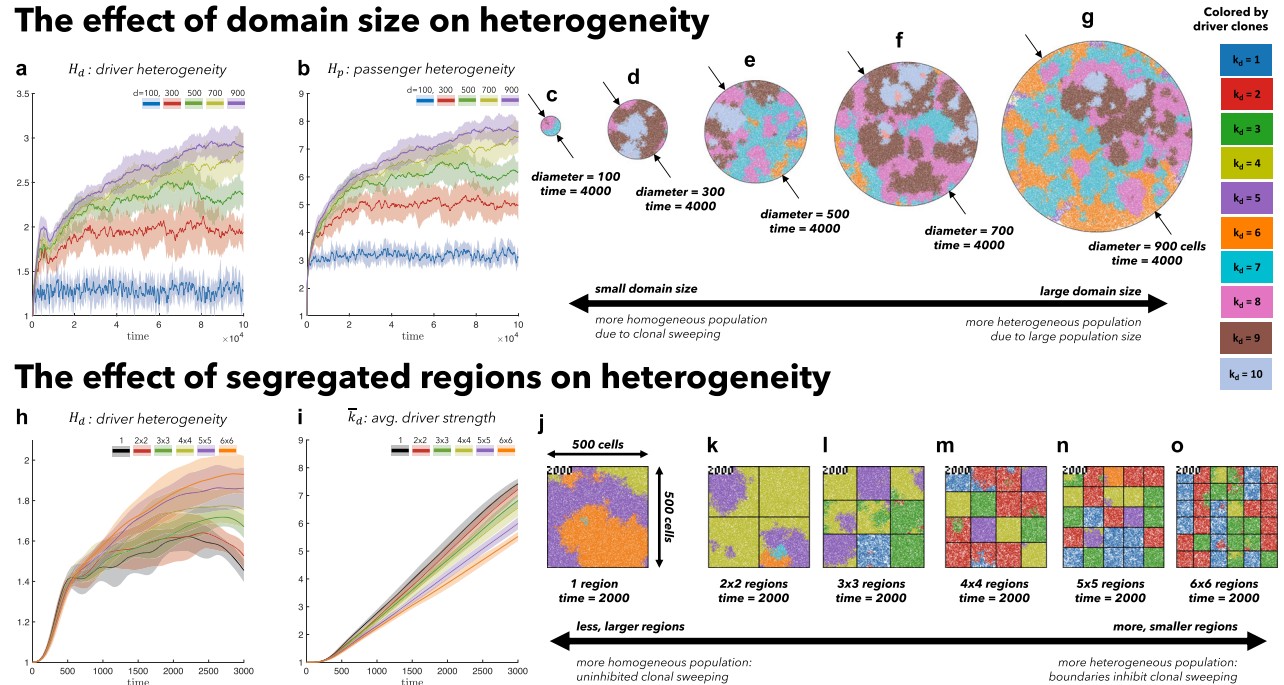

**Fig. 1 The effect of spatial constraints on heterogeneity.** Cells divide and die on a regular square lattice. A cell selected for birth can divide only into an empty grid location and may accrue passenger or driver mutations. Top: simulations on varied sizes of domains, ranging from 100 cells in diameter to 900 cells, seeded with 100 cells ($k_d = 1$, $k_p = 0$) at time zero ($T_p = 5 \cdot 10^6$, $T_d = 700$, $s_d = 0.1$, $s_p = 0.01$). **a, b** An increase in domain size results in increased driver and passenger heterogeneity, with standard deviation shown (shaded colors) for ten stochastic simulations. **c–g** Representative snapshots after 4000 cell generations. Bottom: Identical domain size (seeded with one-third of the domain filled; $k_d = 1$, $k_p = 0$) segregated into varied number of non-interacting regions ($T_p = 10^6$, $T_d = 700$, $s_d = 0.1$, $s_p = 0.01$, $\mu = 10^{-8}$). **h** driver heterogeneity increases with number of segregated regions. **i** Boundaries limit new clones from expanding beyond a single region, decreasing the average number of drivers in the population. **j–o** Representative snapshots after 2000 cell generations. See attached videos V1 and V2. Solid lines indicate mean value, bands indicate ±1 standard deviation for $N = 10$ simulations in **a**, **b**, **h**, **i**.

**The effect of domain size on functional and genetic heterogeneity.** Tumors constrained to smaller domain sizes (Fig. 1, top; video V1) show consistently lower driver and passenger diversity than for larger domain sizes. Small, tightly-coupled homogeneous populations of cells are able to quickly sweep each successive driver mutation. Larger domains consist of a heterogeneous population; with many more cell divisions, the odds of accruing another driver mutation are increased, but they have little chance of sweeping through the large domain. While differences in heterogeneity measures (Fig. 1a, b) diverge quickly for varied domain sizes, they do not approach steady states until extreme time scales. In short, small domain sizes enable clonal sweeping and low diversity. This intuitive model result is consistent across several alternative diversity metrics (Supplementary Fig. 4, rows), for neutral, nearly-neutral, or non-existent passenger mutations (Supplementary Fig. 4, columns).

To control for population size effects in this evolutionary arms race, identical domains are segregated into non-interacting regions of varying size (Fig. 1, bottom; video V2). Again, smaller (highly constrained) regions are more homogeneous. Any clonal sweep stops at each region's boundaries, resulting in a heterogeneous population of locally homogeneous regions (Fig. 1h). Boundaries limit new clones from expanding beyond a single region, decreasing the average number of drivers in the population (Fig. 1i).

**Functional and genetic heterogeneity with limited dispersal.** Bounded, non-interacting regions play a role in human precancerous lesions, which are often locally constrained to a single gland or a duct. Such glandular or ductal structures allow for limited cellular mixing during premalignant growth, enabling the tumor to explore new (and often less constrained) environments. In Fig. 2, each segregated region may now circulate cells into a neighboring region at a low or high rate of mixing (left and right columns, respectively). This model mimics the structure of precancerous breast lesions, the majority of which likely originate within the terminal ductal lobular units (TDLUs) which are connected through a series of extralobular ducts[43]. Similar to the spatially segregated patches (or habitats) commonly found in ecological models, the structure of mammary lobules provides segregation (i.e., the lobule) with some limited dispersal (through the ductal network).

To show how this segregation-dispersal structure accelerates evolution, we plot tumor evolution on two axes: genetic (mutation burden; x-axis) and phenotypic (average number of driver mutations, $\bar{k}_d$; y-axis). The evolution of an unsegregated tumor is shown in black (Fig. 2c, e), accumulating genetic diversity (left-to-right) over time with a slow accumulation of drivers (bottom-to-top). This state-space diagram allows us to track the accelerated acquisition of drivers in a precancerous population, with respect to tumor size.

The trajectory of a tumor's evolution quantifies the tumor-scale effect of domain size and mixing, over time. Despite seeding simulations with identical parameterization, tumors may evolve in a neutral or Darwinian mode (or on a continuous scale between the two, shown in Fig. 2b), subject to selection imposed by domain size and cellular mixing. "Neutral" tumors acquire drivers at a rate equal to the ratio of drivers to all mutations (Fig. 2c, e; blue arrows). Conversely, "Darwinian" tumors sweep each new driver mutation through the population, resulting in a vertical trajectory (Fig. 2c, e; green arrows). There is a continuum between neutral and Darwinian evolution which is time-

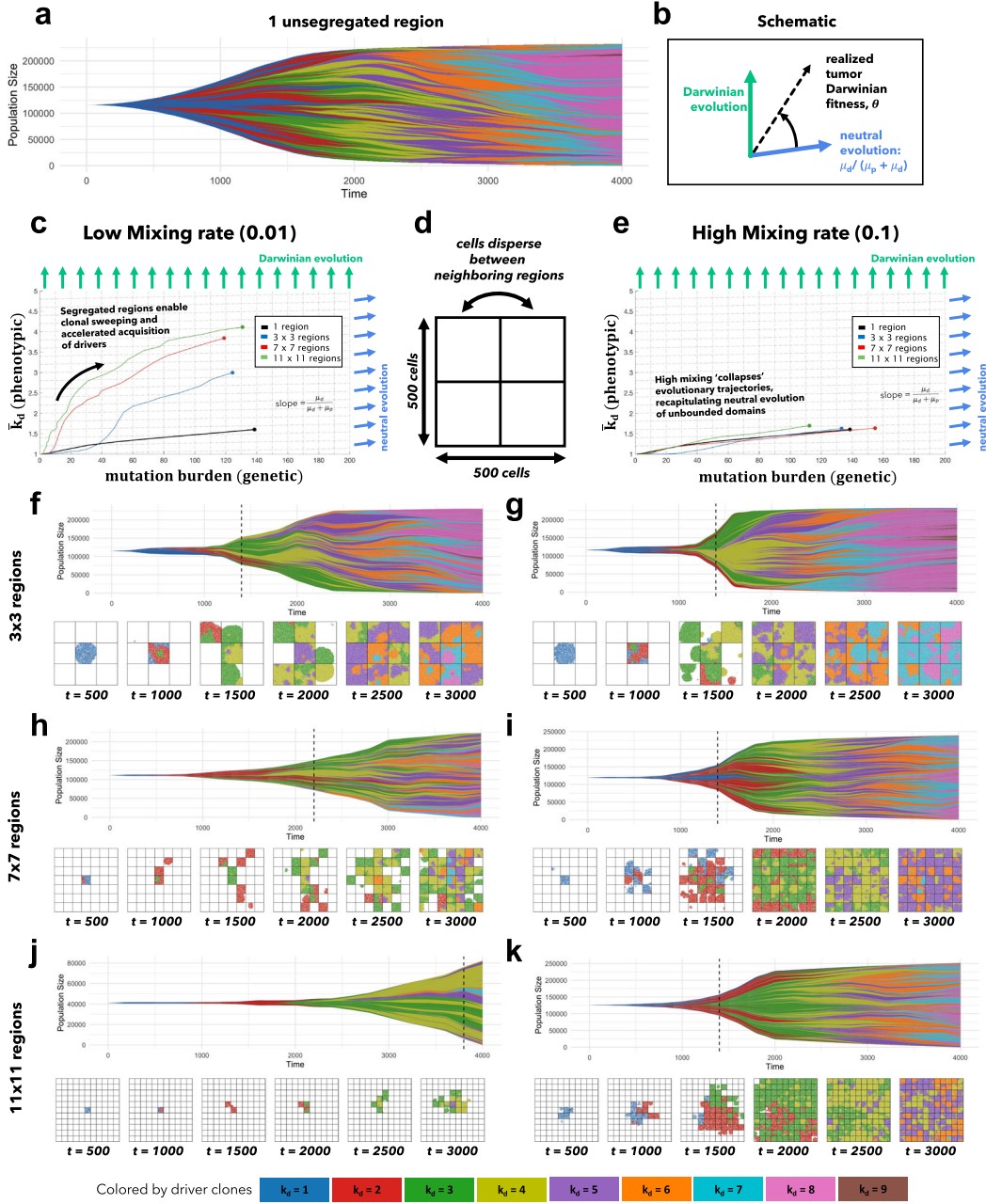

**Fig. 2 Spatial segregation with cell mixing accelerates evolution. a** A Muller plot of tumor evolution represents genotypes color-coded by driver ($k_d$) value. The horizontal axis is time (cell generations), with height corresponding to genotype frequency. Descendant genotypes are shown emerging from inside their parents. **b** A tumor's "realized fitness" can be quantified as the time-varying slope of the evolutionary trajectory. **c** Tumors evolve on the genetic (mutation burden; x-axis) and phenotypic (average drivers; y-axis) axes. For low mixing, smaller regions impose higher selection pressure, accelerating the acquisition of drivers in the population (vertical axis). Simulations are run to identical tumor size (25% of the total domain). **d** The domain is segregated into regions where cells disperse are allowed to mix between segregated regions at a low rate (0.01; **c**; left column) or high rate (0.1; **e** right column). **e** As mixing increases, tumor evolution "collapses" back onto the unsegregated single region case shown in black in (**b**). Snapshots are shown every 500 generations. Simulations repeated for 3 by 3 regions (**f, g**), 7 by 7 regions (**h, i**), and 11 by 11 regions (**j, k**). Parameters: $T_p = 5 \cdot 10^6$, $T_d = 700$, $s_d = 0.1$, $s_p = 10^{-3}$, $\mu = 10^{-8}$. See attached video V3.

dependent, and modulated by the degree of spatial constraints and cellular mixing. The unsegregated tumor traces out a trajectory that lies slightly above the neutral evolution line (Fig. 2c; black), but segregated tumors with limited dispersal trace out more vertical trajectories (Fig. 2c; blue, red, green).

Here we propose a classification of time-dependent tumor evolution as its "realized Darwinian fitness", the time-varying angle on the mutation-driver state space (see Fig. 2b). Tumors are able to realize much higher fitness levels (Fig. 2c) with high spatial constraints and limited mixing, enabling rapid acquisition of drivers, despite no changes in subclonal fitness effects.

This view of Darwinian fitness represents a paradigm shift in tumor neutrality. Previous work has often focused on inferring cell-specific fitness (i.e., subclonal selection) through variant allele frequency distribution metrics (e.g., the 1/f power law distribution)[5,44,45]. Alternatively, we argue that there is a scale

of fitness that should be considered: tumor-scale. The surrounding spatial context modulates the "realized tumor fitness". Spatial constraints accelerate evolution, without changes in cell-specific fitness.

Tumor diversity can be visualized on a Muller plot[46], displaying each clone's abundance over time, colored by driver mutations (Fig. 2f–k; video V3). Tumors with low mixing rates (Fig. 2, left column) tend to evolve by a Darwinian mode of evolution: clonal sweeping which maintains a lower genetic diversity. Increasingly smaller regions allow new drivers to more easily sweep within a local region (Fig. 2c). Introducing segregated regions increases spatial constraints, allowing the tumor to realize increased levels of selection (Fig. 2, left column) even without changes in cell-specific fitness. Importantly, this accelerated mode of evolution is lost when mixing between regions is too high (see Fig. 2e). The high mixing rate (right column) recapitulates the evolutionary trajectory of a relatively unconstrained tumor, decreasing the tumor's realized fitness.

**Spatial context modulates DCIS realized tumor fitness.** As mentioned in the introduction, the seminal study by Casasent et al. performed single-cell DNA sequencing on ten synchronous DCIS-IDC patients to quantify intratumoral heterogeneity while preserving spatial information[20]. Synchronous patients provide an advantage over comparisons of DCIS with recurrent IDC samples which are often collected many years apart. Their findings, reproduced in Fig. 3a–f, indicates a high degree of intratumor heterogeneity within ductal regions with the major clones also present in invasive regions. These data provide evidence for the multiclonal invasion model in DCIS, where one or more clones escape the ducts and migrate into adjacent tissues, maintaining much of the heterogeneity (the IDC heterogeneity is not shown here). This section focuses on the role tissue architecture plays in shaping the pattern of ductal carcinoma heterogeneity.

After DCIS initiation, the branching topology of a breast ductal and glandular network structure acts as an evolutionary accelerant, where spatially segregated regions (ductal branches) work in combination with cell mixing (subject to varied branching topology) to accelerate tumor evolution. Another previously published mathematical model which explicitly modeled ductal structure has indicated that the rate of tumor advance is inversely correlated to the ductal radius, but does not

consider the key role of dispersal and migration into neighboring ducts[47].

Here, the mathematical model is extended to a three-dimensional domain and constrained to grow inside a ductal network reconstructed with data from anthropomorphic breast phantoms[48]. The model is parameterized (see fig. 3g–l; see 'Methods') by performing 10,000 stochastic simulations for a range of driver mutation rates ($\mu_d \in [10^{-7}, 1]$) and fitness ($s_d \in [10^{-3}, 10]$). The simulated evolution of DCIS is initialized and constrained to grow inside a realistic three-dimensional topology of a continuously connected series of progressively smaller branches, as shown in Fig. 4a (see also video V4 and V5, Supplementary Figs. 5 and 6). Measures of clonal heterogeneity for all polyclonal DCIS tumors in ref. [20] were compared to diversity outcomes of the mathematical model (eqn. (3)). The range of parameterizations that recapitulate the heterogeneity for each DCIS tumor is shown in Fig. 3g–l. Parameterizations follow a linear relationship between driver fitness and mutation rate (on a log-scale; characterized by a best-fit to Eqn. (4)). As heterogeneity increases in a–f, the slope of this best fit ($m$) also increases in g–l (see Supplementary Fig. 7).

In Fig. 4, the model is simulated for two patient parameterizations: low (DC13; Fig. 3a, g) and high (DC16; Fig. 3f, l) heterogeneity, subject to different initial spatial conditions. As seen in Fig. 4b, high heterogeneity DC18 (dashed lines) begins with an initial steep Darwinian slope, reaching higher average driver number, albeit with high levels of mutation burden. In contrast, low heterogeneity patient DC13 is characterized by lower slope and lower mutational burden.

There is some evidence that precancerous breast lesions typically originate within lobular areas associated with the highest rates of cellular turnover, especially terminal ductal lobular units (i.e., at the top of Fig. 4a)[43]. Although likely less frequent, we also simulate tumors that originate near the root of the ductal network (i.e., at the bottom of Fig. 4a) to showcase the important role of spatial constraints on tumor evolution. Tumors initiated inside larger ductal branches near the root of the network ($z = 25\%$, blue, Fig. 4b) begin with less spatial constraints and increased access to expand into new branches. These tumors evolve more neutrally (left-to-right trajectories in Fig. 4b). Tumors initiated further from the ductal root in smaller, more constrained branches (e.g., purple curve) are characterized by clonal sweeping

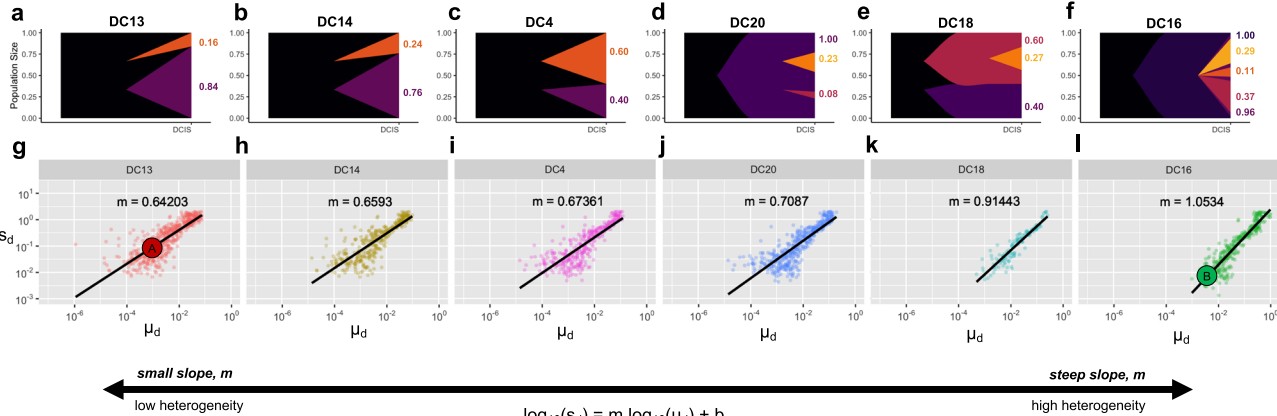

**Fig. 3 Model parameterization of DCIS heterogeneity. a–f** Muller diagrams of inferred clonal heterogeneity of the six polyclonal synchronous DCIS-IDC patients from Casasent et al.[20], arranged from low (**a**) to high (**f**) heterogeneity. **g–l** The mathematical model is extended to a three-dimensional domain and constrained to grow inside a ductal network (Fig. 4a) and 10,000 stochastic simulations for a range of driver mutation rates ($\mu_d \in [10^{-7}, 1]$) and fitness ($s_d \in [10^{-3}, 10]$) are simulated to $N = 10^4$ cells. Parameterizations for which clonal diversity of the mathematical model (eqn. (3)) lie within error bounds reported in Casasent et al. are plotted and fit to eqn. (4). As heterogeneity increases in **a–f**, the slope of this best fit ($m$) also increases in **g–l**. Source data for **a–f** are provided as a Source Data file.

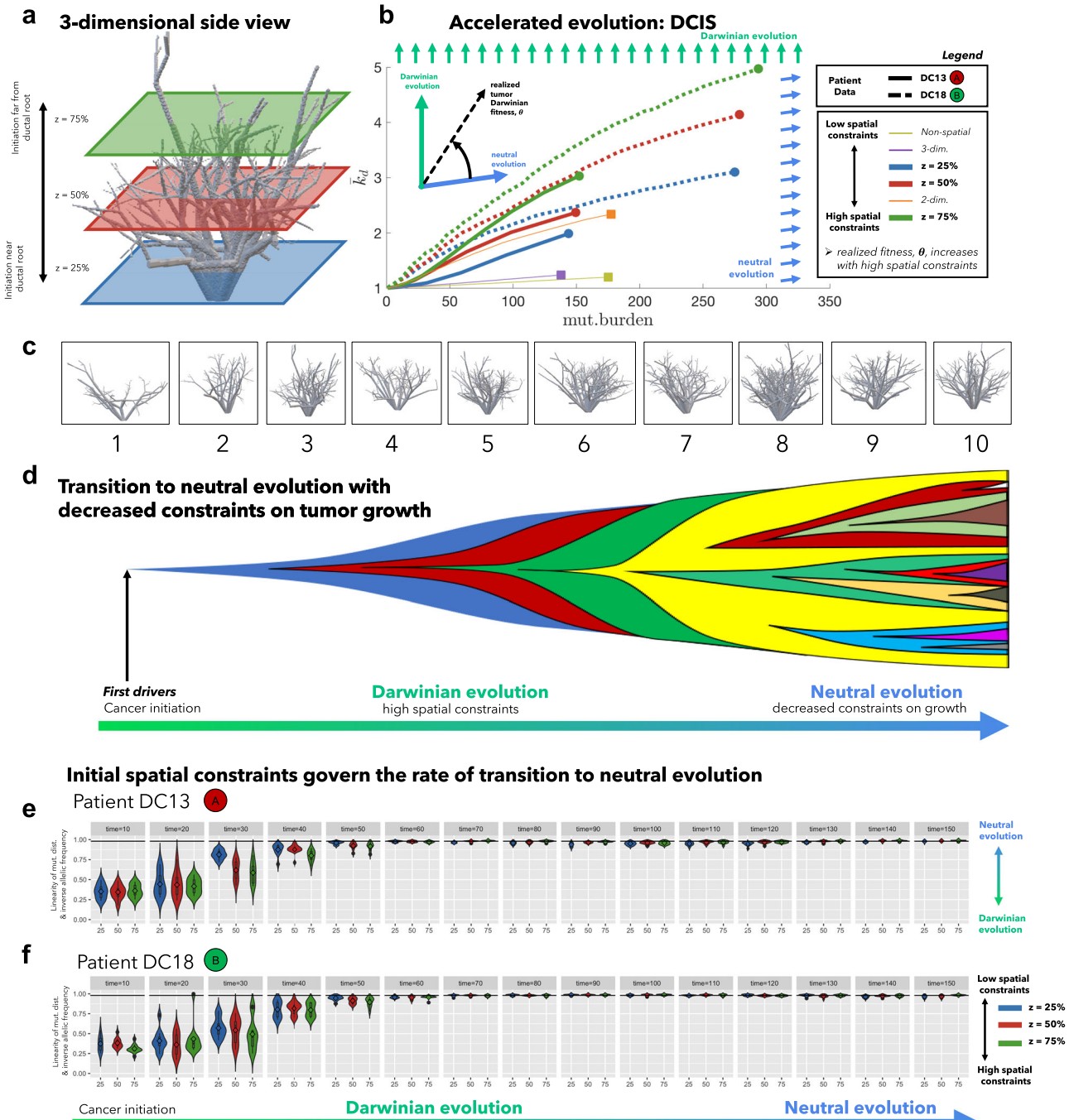

(vertical trajectories) early. At later times, the tumor expands into new unexplored territories, shifting toward neutral trajectories. A tumor originating in a tightly constrained duct enables an accelerated acquisition of drivers early in tumor progression ($10^4$ cells), which may be a more dangerous, highly homogeneous population of malignant cells that have all acquired new traits such as invasiveness, motility, or metastatic capabilities. These important conclusions are lost when considering identical parameterizations of a non-spatial model (Fig. 4b, yellow), unconstrained two-dimensional model (orange), or unconstrained three-dimensional model (purple).

Neutral evolution has previously been quantified using the distribution of the cumulative number of mutations in the tumor with respect to inverse allelic frequency (1/f power-law distribution)[5]. We simulate sequencing (over time) for both patients and quantify the linearity of the 1/f power-law distribution (Fig. 4e, f) and repeat the analysis subject to spatial constraints on ten anthropomorphic breast phantom ductal network reconstructions shown in Fig. 4c. Although all tumors (regardless of spatial constraints) initially are quantitatively shown to be non-neutral (Fig. 4e, f), all eventually progress to a neutral or nearly-neutral evolutionary mode (the neutral minimum threshold of 0.98[5] shown in black). Using this metric, initial high spatial constraints (green bars) transition to neutral evolution more slowly than lack of spatial constraints (blue), especially for patient DC16 (Fig. 4f). When

**Fig. 4 Three-dimensional model of tumor evolution constrained by ductal network structure. a** Realistic three-dimensional topology of breast ductal networks (reconstructed with data from anthropomorphic breast phantoms in[48]) provides full three-dimensional maps to seed and constrain tumor evolution simulations. **b** Tumor evolution is shown for varied points of initiation (z-dimension shown in **a**), to identical sizes ($N = 10^4$ cells). See attached videos V4 and V5. Two parameterizations are shown: DC13 (solid lines; $\mu_p = 10^{-2}$; $\mu_d = 10^{-3.0}$; $s_d = 10^{-1}$; $s_p = 10^{-3}$; indicated as red circle (**a**) in Fig. 3g) and DC16 (solid lines; $\mu_p = 10^{-2}$; $\mu_d = 10^{-2.5}$; $s_d = 10^{-2}$; $s_p = 10^{-3}$; indicated as green circle (**b**) in Fig. 3l). The slope of trajectories (schematic) on this phase portrait is termed "realized tumor Darwinian fitness" and is dependent on spatial constraints at the point of tumor initiation. Simulations closer to the ductal root (e.g., blue curve, $z = 25\%$) in larger, less constrained branches are characterized by a steady left-to-right (neutral) evolution and constant acquisition of new clones. Simulations further from the ductal root (e.g., green, $z = 75\%$) in smaller, more constrained branches are characterized by clonal sweeping (bottom-to-top evolution) early, but with a shift toward neutrality (left-to-right) at later times. Trajectories may be compared to alternative models with identical parameterizations: non-spatial model (dashed yellow), unconstrained two-dimensional model (dashed orange), or unconstrained three-dimensional model (dashed purple). **c** The analysis is repeated subject to spatial constraints on ten highly distinct anthropomorphic breast phantom ductal network reconstructions. In general, ductal branches far from the root decrease in size and increase in number (see Supplementary Fig. 5). **d** All simulations tend to follow an initially Darwinian evolutionary trajectory (steep slope) followed by a transition to neutral evolution (shallow slope). **e, f** An alternative metric of tumor neutrality: the linearity of cumulative mutation distribution with respect to inverse allelic frequency, sometimes called the 1/f power-law distribution[5]. Although all tumors initially are quantitatively shown to be non-neutral, all eventually progress to neutral or nearly-neutral state (the neutral minimum threshold of 0.98[5] shown in black). Tumors with initially high spatial constraints (green violin plots) transition to neutral evolution more slowly than those with less spatial constraints (blue violin plots). High spatial constraints enable accelerated evolution over a range of ductal networks in (**c**) (see Supplementary Fig. 4). Violin plots show median (diamond), 25/75 percentiles, and smallest/largest values within 1.5 times the interquartile range above/below quartiles) for $N = 10$ simulations for each z value.

tumor growth is more spatially constrained it, therefore, consists of a smaller population, enabling easier subclonal expansion of more fit clones (see Supplementary Figs. 8 and 9; Video V6).

## Discussion

Importantly, two otherwise identical tumors may realize dramatic differences in fitness depending on constraints imposed by tissue architecture. On the cell scale, any given subclone may indeed have a selective advantage (i.e., a higher birth rate). Yet, the effective outcome of this subclonal advantage depends on the surrounding competitive context of that cell. In other words, cell-specific phenotypic behavior can be "overridden" by the tissue architecture, allowing the tumor to realize increased fitness. This subclonal selection at the cell-scale may be below the detectable threshold, using traditional metrics, of selection from bulk sequencing methods (i.e., 1/f; Fig. 4e, f). Our approach adds clarity to the debate of neutral tumor evolution by exploring a key mechanism behind both inter-patient and intratumoral tumor heterogeneity: competition for space, in addition to tracking temporal changes.

There is an apparent discrepancy between Figs. 1 and 2. Highest levels of driver acquisition, $k_d$, are found in large domains in Fig. 1 (see panel i). In contrast, the maximal $k_d$ is found in collections of small domains with low mixing in Fig. 2 (see panel c). This illustrates the importance of considering the role of time (Fig. 1), as well as the role of tumor size (Fig. 2) when studying evolution. For example, large domains (see Fig. 2f, h, j) accrue more drivers in equal time, but comparisons of tumors with identical size (Fig. 2f, h, j; vertical dashed lines) reverse this result. The mechanism of increased evolution per cell (or, simply, increased driver acquisition) is as follows: spatial constraints maintain smaller tumors for prolonged periods of time, which facilitate easier clonal sweeps.

It is clear that spatial competition can no longer be ignored in evolutionary models of tumor evolution. These results (summarized in Fig. 5) help to unify the debate surrounding neutral tumor evolution by clarifying the role of space in the transition from Darwinian to neutral evolution. The impact of spatial competition and cellular mixing on cancer evolution (Figs. 1 and 2) is broadly applicable across a range of precancerous lesions, including the particular case study of DCIS shown here.

Initial spatial constraints determine the emergent mode of evolution (neutral to Darwinian) without the necessity for changes in cell-specific mutation rate or fitness effects. The branching topology of ductal networks at tumor initiation determines two important evolutionary accelerants: spatial constraints and cellular mixing. This connectivity is likely to be highly heterogeneous between patients, leading to variability in rates of cellular mixing between spatially distinct niches within a tumor. Although all tumors tend toward neutrality, spatial constraints allow tumors to linger in the non-neutral mode for longer, acquiring more drivers per cell. Limited connectivity enables subclones to undergo high levels of local selection due to spatial constraints. Our metric of realized fitness emphasizes the need to consider both space and time when inferring the mode of evolution. These results indicate that we must be cautious when interpreting non-spatial measures of evolution.

## Methods

Consistent with previous models of passenger driver evolution, tumors will undergo progression with a low mutation rate (less total deleterious passengers) or a low passenger fitness $s_p$ (see Supplementary Fig. 2a, b). We specifically are focused on measures of functional (drivers) and non-functional (passengers) heterogeneity and the conclusions drawn from the model apply in the assumption of neutral or non-neutral passengers.

**Model overview**. Each model simulation is carried out on a two- or three-dimensional grid lattice where each tumor cell is allowed to occupy a single grid point. Simulations are started with $r_0^2$ initial cells ($r_0 = 10$ unless otherwise noted). During each time step, each cell undergoes a birth–death process with the following birth ($P_b$) and death ($P_d$) probabilities:

$$P_b = b\frac{(1 + s_d)^{k_d}}{(1 + s_p)^{k_p}} \tag{1}$$

$$P_d = d \tag{2}$$

where $b$ and $d$ are the baseline birth and death rates, respectively. Tumor cells are initiated with exactly one driver mutation (i.e., $k_d = 1$) and zero passenger mutations (i.e., $k_p = 0$). During the birth process, cells may undergo mutations at a rate $\mu_d = T_d\mu$ (driver mutations) and $\mu_p = T_p\mu$ (passenger mutations). The model is a spatially-explicit extension of ref. [36] where a driver mutation is a rare event with confers a fitness advantage to birth rate known as $s_d$ and a passenger mutation is a relatively common event that confers some fitness penalty, $s_p$. Here we make the simplifying assumption that each subsequent driver (and each subsequent passenger) has an equal effect, rather than a distribution of fitness effects, but others have shown that relaxing this assumption gives similar dynamics[36].

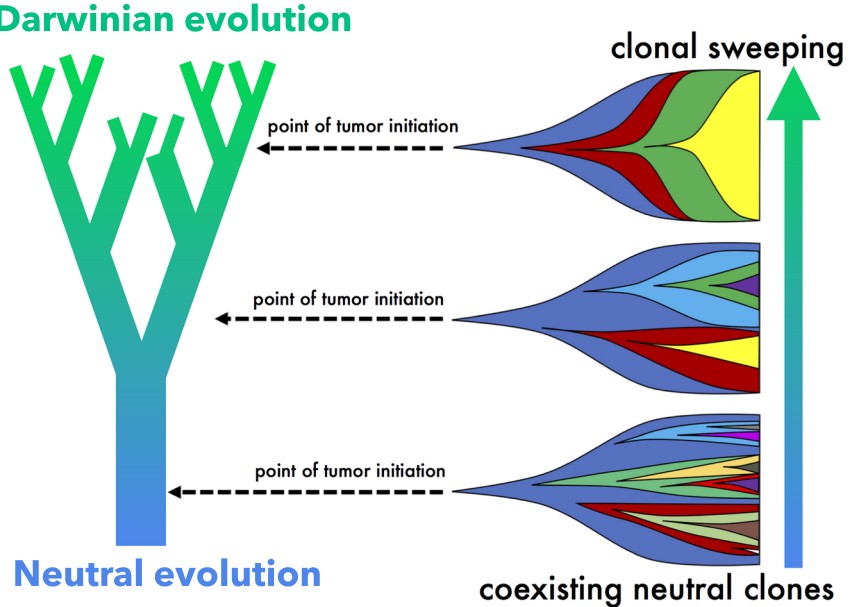

**Fig. 5 Summary schematic.** Simulations near the ductal root in larger, less constrained branches (e.g., bottom) are characterized by a steady, neutral evolution, consistently acquiring new clones that coexist for extended time periods. A smaller tumor originating in a smaller, more constrained branches far from the ductal root (e.g., top) enables an accelerated evolution with clonal sweeping. Levels of selection may be highly heterogeneous between patients, due to the location of tumor initiation.

**Model parameters**. Consistent with the range reported[36,37], the following parameter values were used: $s_d = 0.1$, $s_p = 10^{-3}$, $T_d = 700$, $T_p = 10^6$, $\mu = 10^{-8}$. Due to the possibility of extensive variability in these parameters (see ref. [37]), each of these parameters was varied several orders of magnitude to ensure the robustness of conclusions drawn. Birth and death rates were kept constant at $b = d = 0.5$ unless otherwise noted, and do not significantly alter results for $b, d \in [0.1, 0.5]$ (see Supplementary Fig. 2b).

**Heterogeneity**. Heterogeneity of driver and passenger mutations is calculated using Shannon entropy, given by:

$$H = \exp\left(-\sum_i p_i \log p_i\right) \quad (3)$$

where $p_i$ is the proportion of cells within the population with exactly $i$ driver ($H_d$) or passenger ($H_p$) mutations.

**Parameterization using DCIS patient data**. Model parameterizations were performed by simulating 10,000 stochastic realizations of the mathematical model for a range of driver mutation rates ($\mu_d \in [10^{-8}, 1]$) and fitness ($s_d \in [10^{-3}, 10]$), with constant passenger fitness ($s_p = 10^{-3}$) and rate ($\mu_p = T_p \mu = 10^6 10^{-8} = 10^{-2}$) and spatial location ($z = 75\%$). After the tumor size reaches $10^4$ cells, heterogeneity is measured and plotted in Fig. 3 if it lies within the bounds of error reported in Casasent et al.[20] and a best-fit of all acceptable realizations is performed via the following equation:

$$\log_{10}(s_d) = m \log_{10}(s_d) + b \quad (4)$$

Alternatively, Supplementary Fig. 7 shows the effect of passenger fitness, spatial location, and tumor size on heterogeneity. Note: Casasent et al. uses an alternative definition of Shannon entropy ($-\sum_i (p_i \log p_i)$), which is converted before comparison of heterogeneity in Fig. 3g–l (using Eqn. (3)).

**Dispersal rate between glands and ducts**. Because there are no reliable data on the probability of tumor cell mixing and dispersal between breast ducts, a range of dispersal rates (rate $\in [0.01, 0.1]$) are simulated in dimensions, while realistic breast ductal network structure in three dimensions with varied ductal branch sizes and branching topology (with varied initial conditions) are given by ref. [48].

**Quantification of ductal number and area**. Simulations were performed subject to spatial constraints on ten anthropomorphic breast phantom ductal network reconstructions, shown in Supplementary Fig. 5. For each two-dimensional slice (e.g., Supplementary Fig. 5b), the number of ductal branches is counted and quantified using OpenCV and scikit-image for the Python programming language. In order to minimize bias introduced when ductal branches run parallel to a given slice, an ellipse is fit (Supplementary Fig. 5b, bottom) to each ductal branch to find

the length of the minimum axis, $d_{min}$. Distributions of $d_{min}$ are shown in Supplementary Fig. 5a for each range of $z$ values (colored purple, gray, pink, yellow) and repeated for ten breast ductal network structures. In general, there are fewer, larger ducts near the root of the ductal network and many, smaller ducts as z-layer increases (Supplementary Fig. 5c).

**Reporting summary**. Further information on research design is available in the Nature Research Reporting Summary linked to this article.

## Data availability
No new experimental or clinical data were collected. Data of DCIS-IDC heterogeneity in Fig. 4 may be found at: https://doi.org/10.1016/j.cell.2017.12.007 (ref. [20]). The remaining data and code are available within the Article, Supplementary Information, or available from the authors upon request. Source data are provided with this paper.

## Code availability
All figures were produced using an agent-based modeling platform (Java) known as HAL: Hybrid Automata Library[49]. Ductal branches were counted and quantified using opencv-python (version 4.3.0) and scikit-image (version 0.17.0) for the Python programming language. The code for the model described in this paper is freely available open-source at: https://github.com/MathOnco/tissue-structure-modulates-evolution.

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

## Acknowledgements

The authors gratefully acknowledge funding at the National Cancer Institute via the Physical Sciences Oncology Network (PSON), U54CA193489 (supporting JW, CG, MRT, ARAA), as well as via the Cancer Systems Biology Consortium (CSBC) U01CA232382 (supporting MRT, ARAA), as well as support from the Moffitt Center of Excellence for Evolutionary Therapy (supporting MRT, ARAA). ROS is supported by the Wellcome Trust (grant no. 108861/7/15/7) and the Wellcome Centre for Human Genetics (grant no. 203141/7/16/7). The authors gratefully acknowledge Hanbean Youn for helpful feedback and communication.

## Author contributions

J.W., M.R.-T., and A.R.A.A. conceived the research question and model design. J.W., R.O.S., and C.G. performed the modeling and statistical analyses. All authors contributed to writing and editing the manuscript.

## Competing interests

The authors declare no competing interests.
