## [Peer Review File · Nature Communications]

In the fourth round of peer review Reviewer #1 was unavailable to review the manuscript, therefore Reviewer #5 was recruited to comment on the author's response to Reviewer #1

Reviewers' comments:

Reviewer #1 (Remarks to the Author):

The new focus on premalignant lesions avoids some of the major issues I outlined previously. The most interesting conclusion is that "Cell-specific phenotypic behavior can be "overridden" by the tissue architecture, allowing the tumor to realize increased fitness". However, I still have major reservations about multiple technical aspects of the model and the applicability of the chosen parameters to DCIS. Without running simulations that closely model the specific generation times and population sizes for DCIS and without any attempts to validate the findings in actual DCIS, this new model delivers purely theoretical results without any evidence that this is actually relevant in this premalignant lesions. I am for example surprised that the authors did not comment on my suggestions to try and compare their data to single cell DCIS sequencing results from Navin et al which could have been a first step to reality-check their data.

Furthermore, their revised model is not presented in the context of what is known about the evolution, population sizes, mutation rates or genomic aberration types in DCIS but broadly continues to reference results from evolution studies done on established cancers (see for example the abstract which mainly references studies investigating advanced cancers). This is confusing and makes it difficult to understand what is novel.

There are also problems with the evolutionary terminology: calling some tumors Darwinian and others neutral establishes an unnecessary division as all of these models are evolving based on Darwinian principles.

Major comments on the results:

The authors state that heterogeneity measures do not approach steady states until extreme time scales (20,000 generations; 50 years). Thus, their model assumes a generation time <1 day which is highly questionable for a usually benign lesion such as a DCIS.

The results in figure 1 appear quite surprising when looking at the probabilities of driver mutations based on the model parameters. The simulation is run for 4000 generations with 700 driver loci in the genome and a mutation rate of 10^{-8} . The per driver mutation rate per cell division is therefore only 0.000007. Despite this, most cells have 7-9 drivers at the end of the simulation which seems highly unlikely. The data in figure 1 does also not tally with video 1 where a similar number of drivers is seen in most cells of the population at 9 seconds out of 21 seconds runtime (with 21 seconds representing 100,000 generations). 7-9 driver mutations therefore only seem to appear after ~40000 generations rather than 4000 in a model with the above parameters. What is correct?

Furthermore, it is not clear if such high driver loads are realistic for DCIS. Relevant references are not included, showing a lack of integration with what is known about the biology of DCIS.

“In short, small domain sizes increase the selection pressure enabling fast sweeping and low diversity.”

The authors make this conclusion based on the observation of low diversity and fast sweeping in small domain sizes. However, the number of drivers is higher in many cells in the population with large domain sizes in Figure 1. I would therefore argue that selection pressure is higher and results in a faster increase in fitness in larger domain sizes than in the model with the small size. A sweep is unfortunately a poor surrogate of selection pressure as it heavily depends on populations size.

“Bounded, non-interacting regions play a role in human cancers, which are often locally constrained to a single gland or a duct”

This is one of the statements I specifically criticised in my previous comments. Most advanced cancers are not constrained by ductal or gland structures but show highly chaotic growth. Although the authors say in the rebuttal that they revised the text, the sentence is still unchanged.

Figure 2: I was surprised that the single region always seems to follow a neutral trajectory, particularly given the heterogeneity of drivers observed for a single region in figure 1. This is unfortunately neither explained in the text, nor are representative pictures included in figure 2 to allow readers to assess why there is this big difference. I could also not find any information on the number of cells in this simulation which precluded a more detailed assessment of these results.

“The branching topology of a breast ductal and glandular network structure acts as an evolutionary accelerant, where spatially segregated regions (ductal branches) work in combination with cell mixing (subject to varied branching topology) to accelerate tumor evolution⁴³.”

This sentence suggests that evolution has already been studied in great detail in DCIS which questions what is novel about the authors work. However, looking at reference 43, I could not find any statement about the ductal network acting as an evolutionary accelerant though. This is very confusing.

“Tumors initiated further from the ductal root in smaller, more constrained branches (e.g. purple curve) are characterized by clonal sweeping (vertical trajectories) early. At later times, the tumor expands into new unexplored territories, shifting toward neutral trajectories”

Does this not contradict the overall theory the authors follow which is that spread into new but spatially confined compartment accelerated adaptive evolution?

“Whilst this type of quantification is not currently possible for clinical tissues, our results indicate that we must be cautious when interpreting non-spatial measures of evolution”

This is not correct as resected DCIS lesions should be readily obtainable and Navin et al have shown that these can be analyzed at the single cell level to test for example the hypothesis that populations are more homogenous in small ducts.

Reviewer #3 (Remarks to the Author):

In revising the manuscript and figures, West, Anderson, and colleagues have satisfyingly addressed most of the suggestions and concerns from the initial review. This revised paper more clearly describes their spatial model of tumor progression, offers additional compelling visualizations of the role of spatial constraints and mixing rates on clonal evolution and genetic heterogeneity, and more accurately summarizes several insights regarding realized tumor fitness and the Darwinian-to-neutral spectrum of emergent evolutionary dynamics.

Pages 2 and 4: Both pages contain the sentence "We view this as a paradigm-shifting insight: the surrounding spatial context modulates the 'realized tumor fitness.'" This claim is sweeping but vague, so rather than repeat it twice verbatim, it would be useful to alter at least its second appearance so that it instead precisely defines the quantity referred to as "realized tumor fitness" and clearly identifies which paradigm is being shifted.

Page 8, equation 3: There appears to be an erroneous equals sign.

Reviewer #4 (Remarks to the Author):

This review is focused on the biological accuracy of the spatial model presented in Figure 3 and 4 of the manuscript. While the breast phantom ductal network reconstructions seem to accurately model the branching network of the breast epithelium from larger lactiferous ducts to major ducts and finally the smaller terminal ducts that connect to the lobules, one important aspect that is not integrated in these reconstructions is the actual terminal lobular region of the breast epithelium. The main problem here is that the epithelium in the lobular areas are considered to be the sites with the highest cellular turnover, and the terminal ductal lobular units (TDLUs) represent the most common sites of tumor formation in the human breast both within the lobular ducts as well the lobules themselves (see Honeth et al, 2015, Stem Cell Reports; 10.1016/j.stemcr.2015.02.013; and O'Malley M.B. et al, Breast Pathology. Elsevier, Philadelphia, 2011).

Can the authors explain in more detail what levels of epithelial ducts are being modeled using their breast phantom ductal network reconstructions? In particular, does the size of the ducts in their models reflect lactiferous ducts and major ducts, or would the sizes of terminal ducts be reflected as well?

Considering that terminal ductal lobular units (TDLUs) are the main sites of breast cancer initiation, it would be important to adjust their reconstruction shapes so that the smaller sized terminal ducts and lobular shapes of the breast epithelium are reflected.

Summary of changes to the manuscript for the reviewers:

Summary of the manuscript:

In summary, the manuscript helps to clarify the debate over neutral evolution by exploring a key mechanism which can explain the transition from strong Darwinian effects characteristic of lesions early in tumorigenesis to neutral dynamics of late-stage cancers. This mechanism is the cellular competition for space.

Summary of changes:

We thank the reviewers for the advice to validate the parameterizations used in the model, which we have done in figure 3. Our manuscript is organized as follows: figures 1 and 2 represent a systematic understanding of the mathematical model constrained to various spatial domains. This generalized understanding of the model is subsequently applied to precancerous tumor evolution in figure 4. Parameterizations for figure 4 were determined by confronting the model to data of clonal heterogeneity from DCIS patients in Casasent et. al. *Cell*, 2018, shown in figure 3. As such, figure 4 represents biologically realistic cell numbers and biologically realistic spatial topologies (ductal network structures). Note: the modeling herein does not address late-stage invasive cancers.

Summary of novelty:

In figure 4B we show that the non-spatial modeling commonly used to describe bulk-sequencing data drastically underestimates the propagation of new driver mutations. Intriguingly, even 2- or 3-dimensional models which explicitly consider spatial competition cannot account for this Darwinian-neutral evolutionary transition (see figure 4B). The precise spatial structure of ductal carcinomas strongly influences the competition for space, accelerating evolution by the combination of two factors: spatial constraints and limited cellular mixing. The two factors are subject to the branching topology of the ductal structure, modulating the rate of evolution over time.

The surprising role of spatial constraints is as follows: **without changes in cell-specific fitness, tumors may undergo accelerated acquisition of driver mutations.** The time-dependent evolutionary dynamics are not binary (Darwinian or neutral) but rather continuous.

Brief summary of changes:

While this central message in the manuscript remains unaltered, our updated manuscript represents the following substantial changes in response to reviewer feedback:

1. The key addition in this revision is the parameterizations of the model for all polyclonal DCIS tumors in Casasent et. al., which were compared to diversity outcomes of the mathematical model.
2. Parameterizations (figure 3) follow a linear-relationship between driver fitness and mutation rate (on a log-scale; characterized by a best-fit to eqn. 4. As heterogeneity increases in A-F, the slope of this best fit m also increases in G-L (see figure S6).
3. Figure 4 simulates heterogeneity outcomes for two DCIS patients (DC13 and DC18), showing the transition from Darwinian to neutral evolution for both.

Reviewer #1 (Remarks to the Author):

The new focus on premalignant lesions avoids some of the major issues I outlined previously. The most interesting conclusion is that “Cell-specific phenotypic behavior can be “overridden” by the tissue architecture, allowing the tumor to realize increased fitness”. However, I still have major reservations about multiple technical aspects of the model and the applicability of the chosen parameters to DCIS.

Without running simulations that closely model the specific generation times and population sizes for DCIS and without any attempts to validate the findings in actual DCIS, this new model delivers purely theoretical results without any evidence that this is actually relevant in this premalignant lesions. I am for example surprised that the authors did not comment on my suggestions to try and compare their data to single cell DCIS sequencing results from Navin et al which could have been a first step to reality-check their data.

We thank the reviewer for the suggestion to validate our model findings using Navin et. al. The new version of the manuscript now contains a new figure (figure 3) dedicated to parameterizing the model using data from Navin et. al, as suggested. We have also added a paragraph describing the parameterization procedure, copied below for convenience:

“As mentioned in the introduction, the seminal study by Casasent et. al. performed single-cell DNA sequencing on 10 synchronous DCIS-IDC patients to quantify intratumoral heterogeneity while preserving spatial information. Synchronous patients provide an advantage over comparisons of DCIS with recurrent IDC samples which are often collected many years apart. Their findings, reproduced in figure 4A-F, indicate a high degree of intratumor heterogeneity within ductal regions with the major clones also present in invasive regions. These data provide evidence for the multiclonal invasion model in DCIS, where one or more clones escape the ducts and migrate into adjacent tissues, maintaining much of the heterogeneity (the IDC heterogeneity is not shown here). This section focuses on the role tissue architecture plays in shaping the pattern of ductal carcinoma heterogeneity.

and:

Here, the mathematical model is extended to a three-dimensional domain and constrained to grow inside a ductal network reconstructed with data from anthropomorphic breast phantoms. The model is parameterized (see figure 4G-L; see Methods) by performing by 10,000 stochastic simulations for a range of driver mutation rates ($\mu_d = [10^{-7}, 1]$) and fitness ($s_d = [10^{-3}, 10]$). The simulated evolution of DCIS is initialized and constrained to grow inside a realistic three-dimensional topology of a continuously connected series of progressively smaller branches, as shown in figure 4A (see also video V4 and V5). Measures of clonal heterogeneity for all polyclonal DCIS tumors in Navin et. al. were compared to diversity outcomes of the mathematical model (eqn. 3). The range of parameterizations which recapitulate the heterogeneity for each DCIS tumor are shown in figure 3G-L. Parameterizations follow a linear-relationship between driver fitness and mutation rate (on a log-scale; characterized by a best-fit to eqn. 4. As heterogeneity increases in A-F, the slope of this best fit m also increases in G-L (see figure S6).

Furthermore, their revised model is not presented in the context of what is known about the evolution, population sizes, mutation rates or genomic aberration types in DCIS but broadly continues to reference results from evolution studies done on established cancers (see for example the abstract which mainly references studies investigating advanced cancers). This is confusing and makes it difficult to understand what is novel.

Thank you for the suggestion. We have revised the introduction to include several additional appropriate references. We have clarified the central focus of the manuscript: to understand the role of spatial structure on the **multiclonal invasion model of premalignant heterogeneity**.

We do feel that the results and implications in the manuscript can be applied to various disease cases which rely on a segregation-dispersal structure. However, we have attempted to focus the manuscript results on ductal carcinomas.

There are also problems with the evolutionary terminology: calling some tumors Darwinian and others neutral establishes an unnecessary division as all of these models are evolving based on Darwinian principles.

We appreciate this response, as it is of course true that the simulated tumors are all operating according to Darwinian processes. The main result in figure 4 is two-fold:

- 1) Although cellular interactions are governed by Darwinian processes, the surrounding context of tissue architecture may accelerate these processes
- 2) Although cellular interactions are governed by Darwinian processes, subclonal selection may be below the detectable threshold. This leads to the false assumption that tumors are evolving neutrally, when in reality cell-specific fitness is governed by Darwinian principles.

To highlight these two results, this is the reason we originally introduced this terminology to distinguish between cell-scale Darwinian processes and tumor-scale processes. We labor in the manuscript to explain that cell-specific fitness advantages are held constant.

The authors state that heterogeneity measures do not approach steady states until extreme time scales (20,000 generations; 50 years). Thus, their model assumes a generation time <1 day which is highly questionable for a usually benign lesion such as a DCIS.

The model is agnostic to the length of a generation time in days, so **we have removed this mention**. All plots in figure 1 and 2 reference time in units of 'generations.' Additionally, the key figure in the paper (figure 4B) shows comparisons for identical lesion **size** and not time period.

The results in figure 1 appear quite surprising when looking at the probabilities of driver mutations based on the model parameters. The simulation is run for 4000 generations with 700 driver loci in the genome and a mutation rate of 10^{-8} . The per driver mutation rate per cell division is therefore only 0.000007. Despite this, most cells have 7-9 drivers at the end of the simulation which seems highly unlikely.

We thank the reviewer for the detailed comment. Your description of mutation driver rate is correct (7×10^{-6} , as given by refs 12, 13). The time shown is *generation time*, in which each cell in the population undergoes a birth-death process. Even in a small population domain of 100×100 cells ($\sim 10^4$ cells), the rate of at least one driver event is: $10^4(7 \times 10^{-6})$. This places the rate of each additional driver appearing in the population on the order of 10^{-2} /generation. On average, a new driver event would be expected every 100 generations. This is consistent with findings from MacFarland et. al.

The data in figure 1 does also not tally with video 1 where a similar number of drivers is seen in most cells of the population at 9 seconds out of 21 seconds runtime (with 21 seconds representing 100,000 generations). 7-9 driver mutations therefore only seem to appear after ~ 40000 generations rather than 4000 in a model with the above parameters. What is correct?

We apologize for not providing a timescale to video 1 (the previous version of Video 1 represented ONLY the first 10,000 generations of simulations in figure 1).

We have remedied this by providing a timestep on the video, in the top left corner of each simulation. We have added passenger mutations to the supplemental videos as well. The supplemental Video 1 and 2 now represent the exact simulations shown in figure 1 (top row and bottom row, respectively).

Furthermore, it is not clear if such high driver loads are realistic for DCIS. Relevant references are not included, showing a lack of integration with what is known about the biology of DCIS.

As explained above, parameterizations were done again in light of DCIS measurements from Navin et. al. The simulations in figures 3 and 4 now lie within the clonal heterogeneity reported in Navin et. al (their figure 4B, and 4D).

“In short, small domain sizes increase the selection pressure enabling fast sweeping and low diversity.”

The authors make this conclusion based on the observation of low diversity and fast sweeping in small domain sizes. However, the number of drivers is higher in many cells in the population with large domain sizes in Figure 1. I would therefore argue that selection pressure is higher and results in a faster increase in fitness in larger domain sizes than in the model with the small size. A sweep is unfortunately a poor surrogate of selection pressure as it heavily depends on populations size.

This is a fair critique, and we have re-worded the explanation to focus on heterogeneity (rather than selection).

The reviewer is correct that the highlighted sentence is somewhat ambiguous, so we have updated the wording to the following:

“In short, small domain sizes enable clonal sweeping and low diversity.”

On selection:

- The metric of selection used in the latter half of manuscript is the average number of drivers per cell (k_d). A high selection pressure is indicated by a higher average driver number.
- We agree that a clonal sweep is domain size dependent, which is why we've included figure 1, bottom row for identical domain size. We have re-worded the explanation to focus on heterogeneity (rather than selection).
- As the reviewer intuits, there is indeed an inherent trade-off in population size. A large population undergoes many more birth events, resulting in higher likelihood of new driver mutations. But a large population is more heterogeneous, across a range of metrics (see figure S5).

“Bounded, non-interacting regions play a role in human cancers, which are often locally constrained to a single gland or a duct” This is one of the statements I specifically criticised in my previous comments. Most advanced cancers are not constrained by ductal or gland structures but show highly chaotic growth. Although the authors say in the rebuttal that they revised the text, the sentence is still unchanged.

We apologize for the oversight, and have now changed this sentence to the following:

“Bounded, non-interacting regions play a role in human precancerous lesions, which are often locally constrained to a single gland or a duct.”

Figure 2: I was surprised that the single region always seems to follow a neutral trajectory, particularly given the heterogeneity of drivers observed for a single region in figure 1. This is unfortunately neither

explained in the text, nor are representative pictures included in figure 2 to allow readers to assess why there is this big difference. I could also not find any information on the number of cells in this simulation which precluded a more detailed assessment of these results.

First, it is important to note that the single-region case in figure 2 traces out a trajectory which is slightly above the neutral line of slope $= \mu_d / (\mu_p + \mu_d)$. By our definition, this is a Darwinian (non-neutral) tumor, although very low Darwinian fitness. We have clarified this in the manuscript.

The segregation-dispersal paradigm allows for increased Darwinian fitness (figure 2B), accelerating an already non-neutral trajectory.

We thank the reviewer for the suggestion to add a single-region case to figure 2, and have done so. We have also added the size of the domain (500 by 500, which matches the bottom row of figure 1).

“The branching topology of a breast ductal and glandular network structure acts as an evolutionary accelerant, where spatially segregated regions (ductal branches) work in combination with cell mixing (subject to varied branching topology) to accelerate tumor evolution⁴³.”

This sentence suggests that evolution has already been studied in great detail in DCIS which questions what is novel about the authors work. However, looking at reference 43, I could not find any statement about the ductal network acting as an evolutionary accelerant though. This is very confusing.

The reviewer is correct: the novelty of our study is indeed the ductal network acting as an evolutionary accelerant. The reference links to an alternative mathematical model of DCIS which found that the *rate of advance by the tumor is inversely correlated to ductal radius*, but generally ignores the key role of cellular mixing and dispersal. This is clarified in the text.

“Tumors initiated further from the ductal root in smaller, more constrained branches (e.g. purple curve) are characterized by clonal sweeping (vertical trajectories) early. At later times, the tumor expands into new unexplored territories, shifting toward neutral trajectories”

Does this not contradict the overall theory the authors follow which is that spread into new but spatially confined compartment accelerated adaptive evolution?

No, the lesion is subject to time-varying spatial constraints. Early in time, the lesion is subject to high spatial constraints (low heterogeneity, easy clonal sweeping). Later, cellular dispersal allows for expansion into **less constrained environments**. Larger population sizes also limit expansion rates of new clones, giving rise to neutral evolution at later times. This trend is seen both in controlled domains with low mixing rates (figure 2) and DCIS ductal networks (figure 4), across low/high heterogeneity patients (figure 4B; DC13 and DC18, respectively).

“Whilst this type of quantification is not currently possible for clinical tissues, our results indicate that we must be cautious when interpreting non-spatial measures of evolution”

This is not correct as resected DCIS lesions should be readily obtainable and Navin et al have shown that these can be analyzed at the single cell level to test for example the hypothesis that populations are more homogenous in small ducts.

We thank the reviewer for noting the relevance of Navin et. al. to this manuscript. We have clarified this sentence in the manuscript. We would like to note that the model in our manuscript provides a key insight which Navin’s paper does not: time-dependence of intratumoral heterogeneity.

Reviewer #3 (Remarks to the Author):

In revising the manuscript and figures, West, Anderson, and colleagues have satisfyingly addressed most of the suggestions and concerns from the initial review. This revised paper more clearly describes their spatial model of tumor progression, offers additional compelling visualizations of the role of spatial constraints and mixing rates on clonal evolution and genetic heterogeneity, and more accurately summarizes several insights regarding realized tumor fitness and the Darwinian-to-neutral spectrum of emergent evolutionary dynamics.

Pages 2 and 4: Both pages contain the sentence "We view this as a paradigm-shifting insight: the surrounding spatial context modulates the 'realized tumor fitness.'" This claim is sweeping but vague, so rather than repeat it twice verbatim, it would be useful to alter at least its second appearance so that it instead precisely defines the quantity referred to as "realized tumor fitness" and clearly identifies which paradigm is being shifted.

Thank you for your remarks. We appreciate the reviewer's continued investment in the manuscript.

We have clarified the sentence as suggested, which now reads:

"This leads us to the following insight: the surrounding spatial context modulates the 'realized tumor fitness,' defined as the rate of change of the ratio of driver mutations to mutation burden."

Page 8, equation 3: There appears to be an erroneous equals sign.

Thank you for catching this error! We have removed the erroneous equals sign.

Reviewer #4 (Remarks to the Author):

This review is focused on the biological accuracy of the spatial model presented in Figure 3 and 4 of the manuscript. While the breast phantom ductal network reconstructions seem to accurately model the branching network of the breast epithelium from larger lactiferous ducts to major ducts and finally the smaller terminal ducts that connect to the lobules, one important aspect that is not integrated in these reconstructions is the actual terminal lobular region of the breast epithelium.

The main problem here is that the epithelium in the lobular areas are considered to be the sites with the highest cellular turnover, and the terminal ductal lobular units (TDLUs) represent the most common sites of tumor formation in the human breast both within the lobular ducts as well the lobules themselves (see Honeth et al, 2015, Stem Cell Reports; 10.1016/j.stemcr.2015.02.013; and O'Malley M.B. et al, Breast Pathology. Elsevier, Philadelphia, 2011).

We thank the reviewer for the additional references and the opportunity to clarify the model. The reviewer is correct, the branching network shown in figure 3 represents breast epithelium including major ducts and the smaller terminal ducts. We have updated the manuscript to reflect the fact that the lobular areas are indeed the sites of most common tumor initiation, with a reference to Honeth et. al. (see "Spatial context modulates DCIS realized tumor fitness," paragraph 5).

We invite the reviewer to consider our expanded explanation of figure 2, which we hope will clarify the biological interpretation behind our modeling framework:

"Bounded, non-interacting regions play a role in human precancerous lesions, which are often locally constrained to a single gland or a duct. Such glandular or ductal structures allow for limited cellular mixing during premalignant growth, enabling the tumor to explore new (and often less constrained) environments. In figure 2, each segregated region may now circulate cells into a neighboring region at a low or high rate of mixing (left and right columns, respectively). This model mimics the structure of precancerous breast lesions, the vast majority of which originate within the terminal ductal lobular units (TDLUs) which are connected through a series of extralobular ducts. Similar to the spatially segregated patches (or habitats) commonly found in ecological models, the structure of mammary lobules provides the segregation (i.e. the lobule) with some limited dispersal (through the ductal network)."

It is our point of view that figure 2 can represent a **very simplified** reflection of the effect of lobule size on evolutionary dynamics. For example, larger TDLUs (figure 2D, 2E) acquire drivers more slowly than smaller TDLUs.

After tumor initiation in the TDLU structure, figure 4 addresses a similar but distinct question: the role of the ductal network structure on evolutionary dynamics after initiation in lobules and subsequent growth into terminal ducts. We have added an additional data-driven parameterization section to the manuscript as well:

"As mentioned in the introduction, the seminal study by Casasent et. al. performed single-cell DNA sequencing on 10 synchronous DCIS-IDC patients to quantify intratumoral heterogeneity while preserving spatial information. Synchronous patients provide an advantage over comparisons of DCIS with recurrent IDC samples which are often collected many years apart. Their findings, reproduced in figure 4A-F, indicate a high degree of intratumor heterogeneity within ductal regions with the major clones also present in invasive regions. These data provide evidence for the multiclonal invasion model in DCIS, where one or more clones escape the ducts and migrate into adjacent tissues, maintaining much of the heterogeneity (the IDC heterogeneity is not shown here). This section focuses

on the role tissue architecture plays in shaping the pattern of ductal carcinoma heterogeneity.”

We show that continued evolutionary dynamics depend on the size and branching structure of these terminal ductal structures. We have clarified the language around figures 2 and 4, and would welcome any further clarifying comments from the reviewer.

Can the authors explain in more detail what levels of epithelial ducts are being modeled using their breast phantom ductal network reconstructions? In particular, does the size of the ducts in their models reflect lactiferous ducts and major ducts, or would the sizes of terminal ducts be reflected as well? Considering that terminal ductal lobular units (TDLUs) are the main sites of breast cancer initiation, it would be important to adjust their reconstruction shapes so that the smaller sized terminal ducts and lobular shapes of the breast epithelium are reflected.

Below is a figure from ref. 44 (Jeon, H. et. al. PloS One), where we obtained for the branching topology of ductal networks. The data represents the full ductal network structure, including the terminal ducts. In figure 3, we do not simulate the lobules, but instead focus on the key role that the constrained space that terminal ducts play (after tumor initiation, and growth into the ducts) in accelerating evolutionary dynamics. As explained above, figure 2 can be thought of as a simplified model of evolutionary dynamics within TDLU units, which show that size correlates to the acquisition rate of new drivers.

Fig 4. Generation of glandular tissue distribution.

Fig 6. Modeled breast phantom compartments: (a) external boundary, (b) ductal network, (c) Cooper's ligament, and (d) glandular tissue distribution.

Reviewers' comments:

Reviewer #1 (Remarks to the Author):

I had asked to include evidence and references to show that modelled population sizes and generation numbers are reflecting DCIS lesions. I think this is necessary as a) the model can only be informative if it represents the biological entity with reasonable precision and b) as the authors explicitly state that their model uses realistic cell numbers and timing. Unfortunately, this information is still not there. The authors now included an analysis of data from the Navin lab but it is superficial: It is very hard to understand how the parameters for mutation rate and driver fitness were derived by the insufficiently explained figure 3. In addition, they missed the main opportunity this dataset offers: to assess whether smaller ducts harbour higher or lower numbers of drivers and whether heterogeneity is lower in smaller ducts. They could also have measured the genetic distance between cells in small and large ducts to assess where genetic evolution is fastest.

However, the most important reason for me to remain unconvinced about this manuscript is the authors' reply to this comment I made previously:

"In short, small domain sizes increase the selection pressure enabling fast sweeping and low diversity."

The authors make this conclusion based on the observation of low diversity and fast sweeping in small domain sizes. However, the number of drivers is higher in many cells in the population with large domain sizes in Figure 1. I would therefore argue that selection pressure is higher and results in a faster increase in fitness in larger domain sizes than in the model with the small size. A sweep is unfortunately a poor surrogate of selection pressure as it heavily depends on populations size.

Author Reply:

This is a fair critique, and we have re-worded the explanation to focus on heterogeneity (rather than selection).

The reviewer is correct that the highlighted sentence is somewhat ambiguous, so we have updated the wording to the following:

"In short, small domain sizes enable clonal sweeping and low diversity."

On selection:

- The metric of selection used in the latter half of manuscript is the average number of drivers per cell (kd). A high selection pressure is indicated by a higher average driver number.
- We agree that a clonal sweep is domain size dependent, which is why we've included figure 1, bottom row for identical domain size. We have re-worded the explanation to focus on heterogeneity (rather than selection).
- As the reviewer intuits, there is indeed an inherent trade-off in population size. A large population undergoes many more birth events, resulting in higher likelihood of new driver mutations. But a large population is more heterogeneous, across a range of metrics (see figure S5).

So if they consider the average number of drivers per cell the best metric for selection then:

Figure 1: shows the highest average driver number in the population without constraints (panel J)
Figure 2: shows the highest number of drivers in tumours with no or limited constraints if the same generation time is considered (Muller plots in panels A, F and G). I realize that the authors try to interpret each simulation at different times, when populations size are equal, but allowing the segregated tumors much more time to evolve is a highly questionable approach when assessing the speed of evolution.

Thus, selection according to the definition the authors provide in their rebuttal is HIGHEST in tumours without constraints. This is exactly the opposite of neutral evolution and therefore contradicts the authors' own manuscript title, abstract and conclusions.

Reviewer #4 (Remarks to the Author):

The authors have appropriately addressed my concerns.

Kai Kessenbrock

Reviewer #1 (Remarks to the Author):

I had asked to include evidence and references to show that modelled population sizes and generation numbers are reflecting DCIS lesions. I think this is necessary as a) the model can only be informative if it represents the biological entity with reasonable precision and b) as the authors explicitly state that their model uses realistic cell numbers and timing. Unfortunately, this information is still not there.

On representing the biological entity with reasonable precision:

We have previously addressed the reviewer's concerns about model parameterizations. The model is now directly validated by DCIS evolutionary data (figure 3), via over 20,000 stochastic realizations to recapitulate heterogeneity measured in DCIS patients from Casasent et. al. Cell, 2018.

It is our opinion that this is adequate validation of biological realism given the fact that the conclusions drawn from the model primarily focus on tumor heterogeneity (specifically, the acquisition of driver mutations). The heterogeneity represented in DCIS is demonstrably recapitulated via the model in figure 3.

For further clarification, we will update this claim to focus on the mathematical model's novelty of biologically realistic branching topology:

"We extend these findings on the importance of structure, dispersal, migration, and turnover to a more biologically realistic setting: the 3-dimensional branching topology of a breast ductal network spatial structure, recapitulating the intratumoral heterogeneity in precancerous lesions of ductal carcinoma in situ (DCIS)."

The authors now included an analysis of data from the Navin lab but it is superficial: It is very hard to understand how the parameters for mutation rate and driver fitness were derived by the insufficiently explained figure 3.

We apologize for the insufficient explanation and would welcome further clarification or feedback from the reviewer. We provide a more detailed explanation below.

10,000 stochastic realizations were performed, across a wide range of parameterizations (varied s_d and μ_d). After the tumor is grown to a fixed size ($1e4$ cells), the clonal heterogeneity is measured (Shannon index) and compared to the Shannon index for all DCIS patients in Casasent et. al. If the simulated Shannon lies within error bounds of DCIS-measured Shannon, the parameterization is kept, and plotted in figure 3G through L.

From 3G through L, it is straightforward to see that driver fitness (s_d) and mutation rate (μ_d) are linearly coupled. This novel result shows that increased patient heterogeneity can only be recapitulated by the model with an increased slope of the line describing the relationship between these linearly coupled parameters (called "m" in the manuscript).

To ensure that the final tumor size chosen is not a confounding factor, we've repeated the analysis for several tumor sizes ($1e3$ and $1e4$) in figure S6, which show the same qualitatively trend (high mutation rate results in high heterogeneity).

In addition, they missed the main opportunity this dataset offers: to assess whether smaller ducts harbour higher or lower numbers of drivers and whether heterogeneity is lower in smaller ducts. They could also have measured the genetic distance between cells in small and large ducts to assess where genetic evolution is fastest.

We view this manuscript as primarily addressing the impact of 1) spatial competition and 2) cellular mixing on cancer evolution. The results of figures 1 and 2 are broadly applicable across a range of precancerous lesions (not just DCIS). We apply this broad theme to the particular case study of DCIS by **modeling realistic topology of ductal network architecture**. We wish to note that this is a first – to our knowledge the relationship between DCIS functional and genetic heterogeneity has not been modeled at this scale of network topology. We labor to show in 4B the inadequacy of traditional (non-spatial) methods. Therefore, at this time, we believe that measuring the genetic distance between cells in small or large ducts is outside the reasonable scope of this manuscript.

However, the most important reason for me to remain unconvinced about this manuscript is the authors' reply to this comment I made previously: "In short, small domain sizes increase the selection pressure enabling fast sweeping and low diversity." The authors make this conclusion based on the observation of low diversity and fast sweeping in small domain sizes. However, the number of drivers is higher in many cells in the population with large domain sizes in Figure 1. I would therefore argue that selection pressure is higher and results in a faster increase in fitness in larger domain sizes than in the model with the small size. A sweep is unfortunately a poor surrogate of selection pressure as it heavily depends on populations size. Author Reply:

This is a fair critique, and we have re-worded the explanation to focus on heterogeneity (rather than selection). The reviewer is correct that the highlighted sentence is somewhat ambiguous, so we have updated the wording to the following:

"In short, small domain sizes enable clonal sweeping and low diversity."

On selection:

- *The metric of selection used in the latter half of manuscript is the average number of drivers per cell (kd). A high selection pressure is indicated by a higher average driver number.*
- *We agree that a clonal sweep is domain size dependent, which is why we've included figure 1, bottom row for identical domain size. We have re-worded the explanation to focus on heterogeneity (rather than selection).*
- *As the reviewer intuits, there is indeed an inherent trade-off in population size. A large population undergoes many more birth events, resulting in higher likelihood of new driver mutations. But a large population is more heterogeneous, across a range of metrics (see figure S5).*

So if they consider the average number of drivers per cell the best metric for selection then:

Figure 1: shows the highest average driver number in the population without constraints (panel J)
Figure 2: shows the highest number of drivers in tumours with no or limited constraints if the same generation time is considered (Muller plots in panels A, F and G). I realize that the authors try to interpret each simulation at different times, when populations size are equal, but allowing the segregated tumors much more time to evolve is a highly questionable approach when assessing the speed of evolution.

Thus, selection according to the definition the authors provide in their rebuttal is HIGHEST in tumours without constraints. This is exactly the opposite of neutral evolution and therefore contradicts the authors' own manuscript title, abstract and conclusions.

This review contains a fundamental misunderstanding. In our manuscript, we do not address the speed of evolution. **In fact, the word ‘speed’ never appears** in the text. Although figures 1 and 2 have not significantly changed from initial submission, the above review suggests that these results are contradictory – with no mention of this in the previous two reviews. In fact, the first review had noted that figure 1 appeared to be “intuitively correct.”

We are puzzled at this misunderstanding because previous feedback (in the April 2019 review) that “slow expansion of subclones may simply indicate that selective sweeps are much slower” was addressed by removing any reference to timescale, instead focusing on drivers per cell, not per unit time.

The central result of the manuscript is the apparent “contradiction”:

Figure 1 (bottom) indicates that large domains lead to high driver acquisition (constant time). Figure 2 indicates that *small* segregated domains lead to high driver acquisition (constant tumor size).

Indeed, figure 1 and 2 present opposite conclusions, which is the main focal result of our manuscript! Ignoring the key role of spatial constraints will lead to incorrect conclusions about evolutionary dynamics. The reviewer has stumbled upon the common trap which our manuscript aims to correct.

To be fair, we believe that this fundamental understanding likely derives from our poor word choice of “*acceleration*” in the title. The manuscript is a study of accelerated evolution of drivers per cell, not drivers per time. Again, speed of evolution is never considered, but only acquisition of drivers per tumor cell “accelerated” (or, put simply, increased) by spatial constraints. Given this fundamental confusion, we have revised the manuscript title, as well as the manuscript text in key sections and added a new figure to better articulate these points for further clarification. We welcome further input from the reviewer to aid clarity in understanding.

A more direct comparison of spatial configurations:

The manuscript figures 1 and 2 are initialized with **slightly different parameterizations and initial conditions, making a direct comparison difficult**. Below, we have included an “apples-to-apples” analysis (exact parameterizations and initial conditions) for each spatial structure considered: circles (left column), squares without dispersal (middle column) and squares with dispersal (right column). Spatial constraints increase from left to right (harsher constraints). Our best measure of selection, the average number of drivers (k_d) is measured twice: (i) identical tumor sizes, and (ii) identical time points (see figure, below).

The figure measures the effect of spatial constraints on k_d . The relationship is shown in each subpanel by a linear trend line fit to data, above.

Top row: average # drivers, measured at identical tumor size, n^* :

1. For circular domains, there is a **flat-line** relationship: spatial constraints have no effect on k_d .
2. For segregated square domains, there is a **slight positive** relationship: spatial constraints have a slight positive effect on k_d .
3. For segregated square domains with dispersal, there is a **strong positive** relationship: spatial constraints have a strong positive effect on k_d .

Bottom row: average # drivers, measured at identical time, t^* :

4. For circular domains, there is a **flat-line** relationship: spatial constraints have no effect on k_d .
5. For segregated square domains, there is a **slight negative** relationship: spatial constraints have a slight negative effect on k_d .

6. For segregated square domains with dispersal, there is a **strong negative** relationship: spatial constraints have a strong negative effect on k_d .

This new supplementary figure clearly shows that dispersal is required for **spatial constraints to increase positive selection for driver mutations**. The key result of our manuscript is confirmed: Drivers are increased for simulations with dispersal (green box), relative to the no dispersal baseline in blue. This effect continues to increase with harsh spatial constraints (positive slope).

Contrast this to the bottom row, where we perform the same analysis at a specified time, $t^*=1000$ generations. The effect is the opposite: highly constrained tumors grow more slowly, leading to lower driver numbers.

Our manuscript clearly shows how spatial constraints can push tumors, with identical parameterizations, into a range of emergent modes of evolution (Darwinian to neutral), governed by selection pressure acting on new driver clones.

We strongly believe that using tumors of the same size is a better metric of evolution because this can be (approximately) measured experimentally in mouse models or patients. Within patients, it is difficult to determine the time period over which evolution occurs. **Regardless, our manuscript makes clear that the role of spatial competition on selection pressure within tumors cannot be ignored in either case.**

The mechanism of *'accelerated'* evolution (or, put simply, *increased*) is as follows: spatial constraints maintain smaller tumors for prolonged periods of time, which facilitate clonal sweeps.

We have added some clarification centered around this apparent contradiction to the Discussion section:

“There is an apparent discrepancy between figures 1 and 2. Highest levels of driver acquisition, k_d , are found in large domains in figure 1 (see panel I). In contrast, the maximal k_d is found in collections of small domains with low mixing in figure 2 (see panel C). This illustrates the importance of considering the role of time (figure 1) as well as the role of tumor size (figure 2) when studying evolution. For example, large domains (see figure 2F,H,J) accrue more drivers in equal time, but comparisons of tumors with identical size (figure 2F,H,J; vertical dashed lines) reverses this result. The mechanism of increased evolution per cell (or, simply, increased driver acquisition) is as follows: spatial constraints maintain smaller tumors for prolonged periods of time, which facilitate easier clonal sweeps.”

We feel that this additional supplementary figure clarifies the concerns of the reviewer, and are thankful for their continued advice which has continually made the manuscript stronger.

Reviewer #4 (Remarks to the Author):

The authors have appropriately addressed my concerns.

Kai Kessenbrock

We thank the reviewer for their investment in the manuscript and their helpful suggestions throughout the process.

Reviewer #1 (Remarks to the Author):

I had asked to include evidence and references to show that modelled population sizes and generation numbers are reflecting DCIS lesions. I think this is necessary as a) the model can only be informative if it represents the biological entity with reasonable precision and b) as the authors explicitly state that their model uses realistic cell numbers and timing. Unfortunately, this information is still not there.

On representing the biological entity with reasonable precision:

We have previously addressed the reviewer's concerns about model parameterizations. The model is now directly validated by DCIS evolutionary data (figure 3), via over 20,000 stochastic realizations to recapitulate heterogeneity measured in DCIS patients from Casasent et. al. Cell, 2018.

It is our opinion that this is adequate validation of biological realism given the fact that the conclusions drawn from the model primarily focus on tumor heterogeneity (specifically, the acquisition of driver mutations). The heterogeneity represented in DCIS is demonstrably recapitulated via the model in figure 3.

For further clarification, we will update this claim to focus on the mathematical model's novelty of biologically realistic branching topology:

“We extend these findings on the importance of structure, dispersal, migration, and turnover to a more biologically realistic setting: the 3-dimensional branching topology of a breast ductal network spatial structure, recapitulating the intratumoral heterogeneity in precancerous lesions of ductal carcinoma in situ (DCIS).”

I think here the reviewer asks whether the simulation models the size of a real DCIS lesion, which I guess is in the order of some billions of cells? Therefore, in 2D a section of a lesion would be “circle” of $\sim 1.2M$ cells. I do appreciate that simulating 10,000 instances of large spatial models would take a long time, I think it could be sufficient just to show how the heterogeneity would scale in a few instances of a larger simulation with some 100k or 1M cells. Regarding the generation times, nobody knows and reliable estimates are hard to have, usual cell cycle time varies from 1 to 7 days. A 4 days generation time has been used in the past, but the reviewer may not request to use precise estimates, as there are none.

The authors now included an analysis of data from the Navin lab but it is superficial: It is very hard to understand how the parameters for mutation rate and driver fitness were derived by the insufficiently explained figure 3.

We apologize for the insufficient explanation and would welcome further clarification or feedback from the reviewer. We provide a more detailed explanation below.

10,000 stochastic realizations were performed, across a wide range of parameterizations (varied s_d and μ_d). After the tumor is grown to a fixed size ($1e4$ cells), the clonal heterogeneity is measured (Shannon index) and compared to the Shannon index for all DCIS patients in Casasent et. al. If the simulated Shannon lies within error bounds of DCIS-measured Shannon, the parameterization is kept, and plotted in figure 3G through L.

From 3G through L, it is straightforward to see that driver fitness (s_d) and mutation rate (μ_d) are linearly coupled. This novel result shows that increased patient heterogeneity can only be recapitulated by the model with an increased slope of the line describing the relationship between these linearly coupled parameters (called “m” in the manuscript).

To ensure that the final tumor size chosen is not a confounding factor, we've repeated the analysis for several tumor sizes ($1e3$ and $1e4$) in figure S6, which show the same qualitatively trend (high mutation rate results in high heterogeneity).

Here the authors use the Shannon Index as a summary statistic, which is OK. They infer parameters in a way that is commonly used. Size and sampling may have an impact here and in theory inference should be done on the same conditions, but I do appreciate it's not feasible to run 10k large spatial simulations.

In addition, they missed the main opportunity this dataset offers: to assess whether smaller ducts harbour higher or lower numbers of drivers and whether heterogeneity is lower in smaller ducts. They could also have measured the genetic distance between cells in small and large ducts to assess where genetic evolution is fastest.

We view this manuscript as primarily addressing the impact of 1) spatial competition and 2) cellular mixing on cancer evolution. The results of figures 1 and 2 are broadly applicable across a range of precancerous lesions (not just DCIS). We apply this broad theme to the particular case study of DCIS by **modeling realistic topology of ductal network architecture**. We wish to note that this is a first – to our knowledge the relationship between DCIS functional and genetic heterogeneity has not been modeled at this scale of network topology. We labor to show in 4B the inadequacy of traditional (non-spatial) methods. Therefore, at this time, we believe that measuring the genetic distance between cells in small or large ducts is outside the reasonable scope of this manuscript.

I guess if the data are available, the authors could measure heterogeneity in individual ducts, of course only if the data are annotated appropriately, are they? On the driver events issue, it's much harder, as in these data the original authors just report any CNV that falls in a cancer driver gene annotated by cosmic. That is not how one should identify drivers (there are X COSMIC drivers in chr1 but a gain of one copy of chr1 does not imply one has X number of drivers in the cancer). CNVs, unless an LOH event associated to a mutation or a focal amplification (of several copies), are not generally drivers. Hence almost all annotated drivers in this dataset are not drivers.

However, the most important reason for me to remain unconvinced about this manuscript is the authors' reply to this comment I made previously: "In short, small domain sizes increase the selection pressure enabling fast sweeping and low diversity." The authors make this conclusion based on the observation of low diversity and fast sweeping in small domain sizes.

Yes, just because time to sweep for a given s is smaller in small N .

However, the number of drivers is higher in many cells in the population with large domain sizes in Figure 1.

Yes, just because there are more cells.

I would therefore argue that selection pressure is higher and results in a faster increase in fitness in larger domain sizes than in the model with the small size.

I think it's clear that "faster accumulation of drivers" here is just driven by population size, as there are larger clones with 1 driver in the unconstrained case, it's easier to get additional driver mutations. It's not

driven by selection and competition, but by mutation and population size. This would be true also for passenger mutations with all the same $s=1$.

A sweep is unfortunately a poor surrogate of selection pressure as it heavily depends on populations size.

I think both definition of 'selection pressure' used by authors and reviewers depend on population size.

Author Reply:

This is a fair critique, and we have re-worded the explanation to focus on heterogeneity (rather than selection). The reviewer is correct that the highlighted sentence is somewhat ambiguous, so we have updated the wording to the following:

"In short, small domain sizes enable clonal sweeping and low diversity."

On selection:

- The metric of selection used in the latter half of manuscript is the average number of drivers per cell (kd). A high selection pressure is indicated by a higher average driver number.*
- We agree that a clonal sweep is domain size dependent, which is why we've included figure 1, bottom row for identical domain size. We have re-worded the explanation to focus on heterogeneity (rather than selection).*
- As the reviewer intuit, there is indeed an inherent trade-off in population size. A large population undergoes many more birth events, resulting in higher likelihood of new driver mutations. But a large population is more heterogeneous, across a range of metrics (see figure S5).*

I think believe it is true now what the authors say in the revised wording, that smaller domain size allows for more driver heterogeneity.

So if they consider the average number of drivers per cell the best metric for selection then:

Figure 1: shows the highest average driver number in the population without constraints (panel J)

Figure 2: shows the highest number of drivers in tumours with no or limited constraints if the same generation time is considered (Muller plots in panels A, F and G). I realize that the authors try to interpret each simulation at different times, when populations size are equal, but allowing the segregated tumors much more time to evolve is a highly questionable approach when assessing the speed of evolution.

Thus, selection according to the definition the authors provide in their rebuttal is HIGHEST in tumours without constraints. This is exactly the opposite of neutral evolution and therefore contradicts the authors' own manuscript title, abstract and conclusions.

I think what the authors are trying to state is that in a large, exponentially expanding population without constraints, as there's not much competition, clones with high fitness do not outcompete significantly (or a tall) the other clones, hence the dynamics are effectively neutral. In a large cancer with 1 billion cells, being a new clone with $s=1.2$ won't make any difference. In a small domain, being a new clone with $s=1.2$ will lead to the entire domain being replaced by the new mutant.

Possibly the authors could resolve this discussion by looking at 'significant' clonal expansions / sweeps, and state that they observe a lot of those in small domains, but very few in large domains, as one expects. What is most relevant biologically is large subclones, since small subclones will never become large enough to be detected in an expanding population before the patient's death.

Moving towards a discussion of 'driver heterogeneity' is OK in my opinion. Staying away from hazy definitions of "pressure" and "speed" is helpful.

This review contains a fundamental misunderstanding. In our manuscript, we do not address the speed of evolution. **In fact, the word ‘speed’ never appears** in the text. Although figures 1 and 2 have not significantly changed from initial submission, the above review suggests that these results as contradictory – with no mention of this in the previous two reviews. In fact, the first review had noted that figure 1 appeared to be “intuitively correct.”

We are puzzled at this misunderstanding because previous feedback (in the April 2019 review) that “slow expansion of subclones may simply indicate that selective sweeps are much slower” was addressed by removing any reference to timescale, instead focusing on drivers per cell, not per unit time.

The central result of the manuscript is the apparent “contradiction”:

Figure 1 (bottom) indicates that large domains lead to high driver acquisition (constant time). Figure 2 indicates that *small* segregated domains lead to high driver acquisition (constant tumor size).

Indeed, figure 1 and 2 present opposite conclusions, which is the main focal result of our manuscript! Ignoring the key role of spatial constraints will lead to incorrect conclusions about evolutionary dynamics. The reviewer has stumbled upon the common trap which our manuscript aims to correct.

To be fair, we believe that this fundamental understanding likely derives from our poor word choice of “*acceleration*” in the title. The manuscript is a study of accelerated evolution of drivers per cell, not drivers per time. Again, speed of evolution is never considered, but only acquisition of drivers per tumor cell “accelerated” (or, put simply, increased) by spatial constraints. Given this fundamental confusion, we have revised the manuscript title, as well as the manuscript text in key sections and added a new figure to better articulate these points for further clarification. We welcome further input from the reviewer to aid clarity in understanding.

A more direct comparison of spatial configurations:

The manuscript figures 1 and 2 are initialized with **slightly different parameterizations and initial conditions, making a direct comparison difficult**. Below, we have included an “apples-to-apples” analysis (exact parameterizations and initial conditions) for each spatial structure considered: circles (left column), squares without dispersal (middle column) and squares with dispersal (right column). Spatial constraints increase from left to right (harsher constraints). Our best measure of selection, the average number of drivers (k_d) is measured twice: (i) identical tumor sizes, and (ii) identical time points (see figure, below).

The figure measures the effect of spatial constraints on k_d . The relationship is shown in each subpanel by a linear trend line fit to data, above.

Top row: average # drivers, measured at identical tumor size, n^* :

1. For circular domains, there is a **flat-line** relationship: spatial constraints have no effect on k_d .
2. For segregated square domains, there is a **slight positive** relationship: spatial constraints have a slight positive effect on k_d .
3. For segregated square domains with dispersal, there is a **strong positive** relationship: spatial constraints have a strong positive effect on k_d .

Bottom row: average # drivers, measured at identical time, t^* :

4. For circular domains, there is a **flat-line** relationship: spatial constraints have no effect on k_d .

5. For segregated square domains, there is a **slight negative** relationship: spatial constraints have a slight negative effect on k_d .
6. For segregated square domains with dispersal, there is a **strong negative** relationship: spatial constraints have a strong negative effect on k_d .

This new supplementary figure clearly shows that dispersal is required for **spatial constraints to increase positive selection for driver mutations**. The key result of our manuscript is confirmed: Drivers are increased for simulations with dispersal (green box), relative to the no dispersal baseline in blue. This effect continues to increase with harsh spatial constraints (positive slope).

Contrast this to the bottom row, where we perform the same analysis at a specified time, $t^*=1000$ generations. The effect is the opposite: highly constrained tumors grow more slowly, leading to lower driver numbers.

Our manuscript clearly shows how spatial constraints can push tumors, with identical parameterizations, into a range of emergent modes of evolution (Darwinian to neutral), governed by selection pressure acting on new driver clones.

We strongly believe that using tumors of the same size is a better metric of evolution because this can be (approximately) measured experimentally in mouse models or patients. Within patients, it is difficult to determine the time period over which evolution occurs. **Regardless, our manuscript makes clear that the role of spatial competition on selection pressure within tumors cannot be ignored in either case.**

The mechanism of '*accelerated*' evolution (or, put simply, *increased*) is as follows: spatial constraints maintain smaller tumors for prolonged periods of time, which facilitate clonal sweeps.

We have added some clarification centered around this apparent contradiction to the Discussion section:

“There is an apparent discrepancy between figures 1 and 2. Highest levels of driver acquisition, k_d , are found in large domains in figure 1 (see panel I). In contrast, the maximal k_d is found in collections of small domains with low mixing in figure 2 (see panel C). This illustrates the importance of considering the role of time (figure 1) as well as the role of tumor size (figure 2) when studying evolution. For example, large domains (see figure 2F,H,J) accrue more drivers in equal time, but comparisons of tumors with identical size (figure 2F,H,J; vertical dashed lines) reverses this result. The mechanism of increased evolution per cell (or, simply, increased driver acquisition) is as follows: spatial constraints maintain smaller tumors for prolonged periods of time, which facilitate easier clonal sweeps.”

We feel that this additional supplementary figure clarifies the concerns of the reviewer, and are thankful for their continued advice which has continually made the manuscript stronger.

Reviewer #4 (Remarks to the Author):

The authors have appropriately addressed my concerns.

Kai Kessenbrock

We thank the reviewer for their investment in the manuscript and their helpful suggestions throughout the process.

Color-coding legend:

- **BLACK text: Reviewer #1's original review**
- **BLUE text: author initial response**
- **RED text: Reviewer #5's review of our author's response to #1**
- **GREEN text: author response to #5**

Summary of changes to the manuscript:

In summary, Reviewer #5's response can be grouped into 3 broad categories:

1. Simulating realistic cell sizes

The reviewer's first response deals with realistic cell size. Reviewer 5 has suggested showing how heterogeneity scales up to ~1.2 million cells. This was easily addressable, and we've included a new supplementary figure (*Supplemental Figure S1*).

2. Validation of model via Casasent et. al. dataset.

We note that we have used this dataset to validate the model by matching the summary statistic in the data (Shannon diversity) to model output. Reviewer #5 agrees that this method is reasonable.

While Reviewer #1 suggests *further* validation by this dataset (Casasent, 2018), Reviewer #5 suggests several issues with this approach. We have provided expanded discussion on the issues and challenges of validating our model with this dataset (in agreement w/ reviewer 5).

Given these issues and the disagreement between reviewers we humbly suggest not pursuing reviewer #1's original request any further.

3. Significant clonal expansion

Reviewer #5 provides an insightful suggestion to look at 'significant' clonal expansions to quantify the effect of domain size on clonal sweeping. This is easily addressable, and we've included a new supplementary figure and video (*Supplemental Figure S9, Supplemental Video V6*).

We have also prepared point-by-point response to each point below.

Reviewer #1 (Remarks to the Author):

I had asked to include evidence and references to show that modelled population sizes and generation numbers are reflecting DCIS lesions. I think this is necessary as a) the model can only be informative if it represents the biological entity with reasonable precision and b) as the authors explicitly state that their model uses realistic cell numbers and timing. Unfortunately, this information is still not there.

On representing the biological entity with reasonable precision:

We have previously addressed the reviewer's concerns about model parameterizations. The model is now directly validated by DCIS evolutionary data (figure 3), via over 20,000 stochastic realizations to recapitulate heterogeneity measured in DCIS patients from Casasent et. al. Cell, 2018.

It is our opinion that this is adequate validation of biological realism given the fact that the conclusions drawn from the model primarily focus on tumor heterogeneity (specifically, the acquisition of driver mutations). The heterogeneity represented in DCIS is demonstrably recapitulated via the model in figure 3.

For further clarification, we will update this claim to focus on the mathematical model's novelty of biologically realistic branching topology:

"We extend these findings on the importance of structure, dispersal, migration, and turnover to a more biologically realistic setting: the 3-dimensional branching topology of a breast ductal network spatial structure, recapitulating the intratumoral heterogeneity in precancerous lesions of ductal carcinoma in situ (DCIS)."

I think here the reviewer asks whether the simulation models the size of a real DCIS lesion, which I guess is in the order of some billions of cells? Therefore, in 2D a section of a lesion would be "circle" of ~1.2M cells. I do appreciate that simulating 10,000 instances of large spatial models would take a long time, I think it could be sufficient just to show how the heterogeneity would scale in a few instances of a larger simulation with some 100k or 1M cells.

Previously, the 2d cross sections were ~600,000 cells, and we have now scaled to include ~1.5 million cells (circular domain with 1500 cells in diameter). We've repeated the analysis in figure 1 (top), to indicate how heterogeneity scales with more realistic cell numbers (see figure A, B, C below). This is now supplemental figure S1 in the manuscript.

The driver and passenger heterogeneity (Shannon) at the final time point (100,000 generations) is shown in D, E, respectively. Heterogeneity increases with increased domain size. The heterogeneity begins to taper off at larger domain sizes because heterogeneity has not reached steady state (see A,B) even at extreme time scales).

The effect of domain size on heterogeneity

Regarding the generation times, nobody knows and reliable estimates are hard to have, usual cell cycle time varies from 1 to 7 days. A 4 days generation time has been used in the past, but the reviewer may not request to use precise estimates, as there are none.

We agree with the Reviewer's assessment of issues converting generation times & doubling times. To deal with this, figures 1 and 2 report time in units of "generations" (not days), directly obtained from our mathematical modeling results. Precise estimates of conversion from generations to days is not necessary for the main results in the manuscript.

The authors now included an analysis of data from the Navin lab but it is superficial: It is very hard to understand how the parameters for mutation rate and driver fitness were derived by the insufficiently explained figure 3.

We apologize for the insufficient explanation and would welcome further clarification or feedback from the reviewer. We provide a more detailed explanation below.

10,000 stochastic realizations were performed, across a wide range of parameterizations (varied s_d and μ_d). After the tumor is grown to a fixed size ($1e4$ cells), the clonal heterogeneity is measured (Shannon index) and compared to the Shannon index for all DCIS patients in Casant et. al. If the simulated Shannon lies within error bounds of DCIS-measured Shannon, the parameterization is kept, and plotted in figure 3G through L.

From 3G through L, it is straightforward to see that driver fitness (s_d) and mutation rate (μ_d) are linearly coupled. This novel result shows that increased patient heterogeneity can only be recapitulated by the

model with an increased slope of the line describing the relationship between these linearly coupled parameters (called “m” in the manuscript).

To ensure that the final tumor size chosen is not a confounding factor, we’ve repeated the analysis for several tumor sizes (1e3 and 1e4) in figure S6, which show the same qualitatively trend (high mutation rate results in high heterogeneity).

Here the authors use the Shannon Index as a summary statistic, which is OK. They infer parameters in a way that is commonly used. Size and sampling may have an impact here and in theory inference should be done on the same conditions, but I do appreciate it’s not feasible to run 10k large spatial simulations.

Correct, Shannon Index is used as a summary statistic. Additionally, in figure S7 we have previously provided insight into the effect of tumor size on Shannon diversity.

In addition, they missed the main opportunity this dataset offers: to assess whether smaller ducts harbour higher or lower numbers of drivers and whether heterogeneity is lower in smaller ducts. They could also have measured the genetic distance between cells in small and large ducts to assess where genetic evolution is fastest.

We view this manuscript as primarily addressing the impact of 1) spatial competition and 2) cellular mixing on cancer evolution. The results of figures 1 and 2 are broadly applicable across a range of precancerous lesions (not just DCIS). We apply this broad theme to the particular case study of DCIS by **modeling realistic topology of ductal network architecture**. We wish to note that this is a first – to our knowledge the relationship between DCIS functional and genetic heterogeneity has not been modeled at this scale of network topology. We labor to show in 4B the inadequacy of traditional (non-spatial) methods. Therefore, at this time, we believe that measuring the genetic distance between cells in small or large ducts is outside the reasonable scope of this manuscript.

I guess if the data are available, the authors could measure heterogeneity in individual ducts, of course only if the data are annotated appropriately, are they? On the driver events issue, it’s much harder, as in these data the original authors just report any CNV that falls in a cancer driver gene annotated by cosmic. That is not how one should identify drivers (there are X COSMIC drivers in chr1 but a gain of one copy of chr1 does not imply one has X number of drivers in the cancer). CNVs, unless an LOH event associated to a mutation or a focal amplification (of several copies), are not generally drivers. Hence almost all annotated drivers in this dataset are not drivers.

The reviewer is correct, to do this analysis we would need the following annotations within the dataset:

- number of cells per duct (or ductal area to be able to infer total cell count)
- annotation of mutations classified as driver or passenger
 - mutational frequencies with copy number corrections
- number of drivers per duct
- number of passengers per duct

Unfortunately, the Casasent et. al. dataset is not appropriate to draw robust conclusions for measuring the heterogeneity of drivers or passengers within individual ducts. The Casasent et. al. draws inferences between invasive and DCIS regions of synchronous DCIS-IDC patients. Within the manuscript, they report representative samplings from single duct data, but we see no clear method of expanding this to the resolution necessary for inferring heterogeneity based on duct size.

For example, it's challenging to determine accurate cell counts for each duct based on this dataset. However, if serially aligned histologies with LCM of individual ducts was present, this may be possible.

There is an additional challenge. These data report copy-number corrected mutational frequencies at an impressive resolution; however, there is ill-consensus on how structural variations precede and contribute to driver mutational events. We agree with the reviewer's point that additional copies of chromosomes does not imply a 1-to-1 addition of drivers.

For these broad reasons, we continue to maintain that this analysis is outside the scope of our manuscript and likely not possible given the constraints of the dataset.

Our model talks to the dynamics within and cellular mixing between ducts of varied size. In this response, we have not planned to further investigate the Casasent et. al. dataset (at the discretion of the editor).

However, the most important reason for me to remain unconvinced about this manuscript is the authors' reply to this comment I made previously: "In short, small domain sizes increase the selection pressure enabling fast sweeping and low diversity." The authors make this conclusion based on the observation of low diversity and fast sweeping in small domain sizes.

Yes, just because time to sweep for a given s is smaller in small N .

Agreed. We have previously mentioned this in the manuscript with this sentence: "Small, tightly-coupled homogeneous populations of cells are able to quickly sweep each successive driver mutation."

However, the number of drivers is higher in many cells in the population with large domain sizes in Figure 1.

Yes, just because there are more cells.

Agreed, therein lies the tradeoff between more cells (more opportunity for driver mutations), and smaller domain sizes (lower time for clonal sweep).

I would therefore argue that selection pressure is higher and results in a faster increase in fitness in larger domain sizes than in the model with the small size.

I think it's clear that "faster accumulation of drivers" here is just driven by population size, as there are larger clones with 1 driver in the unconstrained case, it's easier to get additional driver mutations. It's not driven by selection and competition, but by mutation and population size. This would be true also for passenger mutations with all the same $s=1$.

Agreed (see full response below).

A sweep is unfortunately a poor surrogate of selection pressure as it heavily depends on populations size.

I think both definition of 'selection pressure' used by authors and reviewers depend on population size.

Agreed (see full response below).

Author Reply:

This is a fair critique, and we have re-worded the explanation to focus on heterogeneity (rather than selection). The reviewer is correct that the highlighted sentence is somewhat ambiguous, so we have updated the wording to the following:

“In short, small domain sizes enable clonal sweeping and low diversity.”

On selection:

- The metric of selection used in the latter half of manuscript is the average number of drivers per cell (kd). A high selection pressure is indicated by a higher average driver number.*
- We agree that a clonal sweep is domain size dependent, which is why we’ve included figure 1, bottom row for identical domain size. We have re-worded the explanation to focus on heterogeneity (rather than selection).*
- As the reviewer intuits, there is indeed an inherent trade-off in population size. A large population undergoes many more birth events, resulting in higher likelihood of new driver mutations. But a large population is more heterogeneous, across a range of metrics (see figure S5).*

I think believe it is true now what the authors say in the revised wording, that smaller domain size allows for more driver heterogeneity.

Figure 1 now focuses on heterogeneity, avoiding some of the pitfalls that both reviewers have mentioned. Please see full response below.

So if they consider the average number of drivers per cell the best metric for selection then:

Figure 1: shows the highest average driver number in the population without constraints (panel J)

Figure 2: shows the highest number of drivers in tumours with no or limited constraints if the same generation time is considered (Muller plots in panels A, F and G). I realize that the authors try to interpret each simulation at different times, when populations size are equal, but allowing the segregated tumors much more time to evolve is a highly questionable approach when assessing the speed of evolution.

Thus, selection according to the definition the authors provide in their rebuttal is HIGHEST in tumours without constraints. This is exactly the opposite of neutral evolution and therefore contradicts the authors’ own manuscript title, abstract and conclusions.

I think what the authors are trying to state is that in a large, exponentially expanding population without constraints, as there’s not much competition, clones with high fitness do not outcompete significantly (or a tall) the other clones, hence the dynamics are effectively neutral. In a large cancer with 1 billion cells, being a new clone with $s=1.2$ won’t make any difference. In a small domain, being a new clone with $s=1.2$ will lead to the entire domain being replaced by the new mutant.

Possibly the authors could resolve this discussion by looking at ‘significant’ clonal expansions / sweeps, and state that they observe a lot of those in small domains, but very few in large domains, as one expects. What is most relevant biologically is large subclones, since small subclones will never become large enough to be detected in an expanding population before the patient’s death.

Thank you for the suggestion, this is an insightful technique to quantify the effect of domain size on clonal sweeping. We’ve added a new supplemental figure (figure S9) showing the fraction of tumor cells belonging to the largest clone at a given time. The smallest domain size (100 cells in diameter) corresponds to the highest fraction. This follows the reviewer’s intuition that clonal sweeps are more likely in small domains.

The analysis is repeated for no passengers (exaggerating the effect), deleterious and neutral passengers. In all three scenarios, smaller domains are more likely to have significant clones with a large tumor fraction.

Note: when passenger mutations are included the domain contains more clones overall, leading to a smaller fraction taken by the largest clone. Yet, in all cases the largest fraction occurs for small domains.

Fraction of tumor taken by largest clone ($t^*=4000$)

We also note that whilst this new metric is important, it is supplemental to the main analysis in the manuscript. Our main metric (average number of drivers) provides a more continuous measure of driver acquisition because it accounts for the current sweeping clone, as well as any additional more fit clones sweeping quickly thereafter.

We have also added Supplemental Video V6 to accompany this figure and analysis. In this video the largest clone at each point in time is colored dark blue, with second largest colored light blue. All other cells are colored dark gray. Visually, smaller domains (left) consist of these two dominant clones, while the fraction of largest clone is smaller for large domains. See the stacked bar chart to the right of each domain for a more relative quantification of the sizes.

Moving towards a discussion of 'driver heterogeneity' is OK in my opinion. Staying away from hazy definitions of "pressure" and "speed" is helpful.

This review contains a fundamental misunderstanding. In our manuscript, we do not address the speed of evolution. **In fact, the word 'speed' never appears** in the text. Although figures 1 and 2 have not significantly changed from initial submission, the above review suggests that these results as contradictory – with no mention of this in the previous two reviews. In fact, the first review had noted that figure 1 appeared to be "intuitively correct."

We are puzzled at this misunderstanding because previous feedback (in the April 2019 review) that “slow expansion of subclones may simply indicate that selective sweeps are much slower” was addressed by removing any reference to timescale, instead focusing on drivers per cell, not per unit time.

The central result of the manuscript is the apparent “contradiction”:

Figure 1 (bottom) indicates that large domains lead to high driver acquisition (constant time). Figure 2 indicates that *small* segregated domains lead to high driver acquisition (constant tumor size).

Indeed, figure 1 and 2 present opposite conclusions, which is the main focal result of our manuscript!

Ignoring the key role of spatial constraints will lead to incorrect conclusions about evolutionary dynamics. The reviewer has stumbled upon the common trap which our manuscript aims to correct.

To be fair, we believe that this fundamental understanding likely derives from our poor word choice of “*acceleration*” in the title. The manuscript is a study of accelerated evolution of drivers per cell, not drivers per time. Again, speed of evolution is never considered, but only acquisition of drivers per tumor cell “accelerated” (or, put simply, increased) by spatial constraints. Given this fundamental confusion, we have revised the manuscript title, as well as the manuscript text in key sections and added a new figure to better articulate these points for further clarification. We welcome further input from the reviewer to aid clarity in understanding.

A more direct comparison of spatial configurations:

The manuscript figures 1 and 2 are initialized with **slightly different parameterizations and initial conditions, making a direct comparison difficult**. Below, we have included an “apples-to-apples” analysis (exact parameterizations and initial conditions) for each spatial structure considered: circles (left column), squares without dispersal (middle column) and squares with dispersal (right column). Spatial constraints increase from left to right (harsher constraints). Our best measure of selection, the average number of drivers (k_d) is measured twice: (i) identical tumor sizes, and (ii) identical time points (see figure, below). The figure measures the effect of spatial constraints on k_d . The relationship is shown in each subpanel by a linear trend line fit to data, above.

Top row: average # drivers, measured at identical tumor size, n^* :

1. For circular domains, there is a **flat-line** relationship: spatial constraints have no effect on k_d .
2. For segregated square domains, there is a **slight positive** relationship: spatial constraints have a slight positive effect on k_d .
3. For segregated square domains with dispersal, there is a **strong positive** relationship: spatial constraints have a strong positive effect on k_d .

Bottom row: average # drivers, measured at identical time, t^* :

4. For circular domains, there is a **flat-line** relationship: spatial constraints have no effect on k_d .
5. For segregated square domains, there is a **slight negative** relationship: spatial constraints have a slight negative effect on k_d .
6. For segregated square domains with dispersal, there is a **strong negative** relationship: spatial constraints have a strong negative effect on k_d .

This new supplementary figure clearly shows that dispersal is required for **spatial constraints to increase positive selection for driver mutations**. The key result of our manuscript is confirmed: Drivers are increased for simulations with dispersal (green box), relative to the no dispersal baseline in blue. This effect continues to increase with harsh spatial constraints (positive slope).

Contrast this to the bottom row, where we perform the same analysis at a specified time, $t^*=1000$ generations. The effect is the opposite: highly constrained tumors grow more slowly, leading to lower driver numbers.

Our manuscript clearly shows how spatial constraints can push tumors, with identical parameterizations, into a range of emergent modes of evolution (Darwinian to neutral), governed by selection pressure acting on new driver clones.

We strongly believe that using tumors of the same size is a better metric of evolution because this can be (approximately) measured experimentally in mouse models or patients. Within patients, it is difficult to determine the time period over which evolution occurs. **Regardless, our manuscript makes clear that the role of spatial competition on selection pressure within tumors cannot be ignored in either case.**

The mechanism of '*accelerated*' evolution (or, put simply, *increased*) is as follows: spatial constraints maintain smaller tumors for prolonged periods of time, which facilitate clonal sweeps.

We have added some clarification centered around this apparent contradiction to the Discussion section:

“There is an apparent discrepancy between figures 1 and 2. Highest levels of driver acquisition, k_d , are found in large domains in figure 1 (see panel I). In contrast, the maximal k_d is found in collections of small domains with low mixing in figure 2 (see panel C). This illustrates the importance of considering the role of time (figure 1) as well as the role of tumor size (figure 2) when studying evolution. For example, large domains (see figure 2F,H,J) accrue more drivers in equal time, but comparisons of tumors with identical size (figure 2F,H,J; vertical dashed lines) reverses this result. The mechanism of increased evolution per cell (or, simply, increased driver acquisition) is as follows: spatial constraints maintain smaller tumors for prolonged periods of time, which facilitate easier clonal sweeps.”

We feel that this additional supplementary figure clarifies the concerns of the reviewer, and are thankful for their continued advice which has continually made the manuscript stronger.

REVIEWERS' COMMENTS

Reviewer #5 (Remarks to the Author):

The authors addressed all my comments.

Point by Point response to reviews:

Reviewer #5 (Remarks to the Author):

The authors addressed all my comments.

We thank the reviewer for their time and attention responding and critiquing our manuscript. It was very beneficial!